# CLARE: Conservative Model-Based Reward Learning for Offline Inverse Reinforcement Learning

**Sheng Yue**[1]*, **Guanbo Wang**[2], **Wei Shao**[3]*, **Zhaofeng Zhang**[4], **Sen Lin**[5]*, **Ju Ren**[1,6]†,
**Junshan Zhang**[3]
[1]Tsinghua University, [2]Tongji University, [3]University of California, Davis,
[4]Arizona State University, [5]Ohio State University, [6]Zhongguancun Laboratory

## Abstract

This work aims to tackle a major challenge in offline Inverse Reinforcement Learning (IRL), namely the *reward extrapolation error*, where the learned reward function may fail to explain the task correctly and misguide the agent in unseen environments due to the intrinsic covariate shift. Leveraging both expert data and lower-quality diverse data, we devise a principled algorithm (namely CLARE) that solves offline IRL efficiently via integrating "conservatism" into a learned reward function and utilizing an estimated dynamics model. Our theoretical analysis provides an upper bound on the return gap between the learned policy and the expert policy, based on which we characterize the impact of covariate shift by examining subtle two-tier tradeoffs between the "exploitation" (on both expert and diverse data) and "exploration" (on the estimated dynamics model). We show that CLARE can provably alleviate the reward extrapolation error by striking the right "exploitation-exploration" balance therein. Extensive experiments corroborate the significant performance gains of CLARE over existing state-of-the-art algorithms on MuJoCo continuous control tasks (especially with a small offline dataset), and the learned reward is highly instructive for further learning (source code).

## 1 Introduction

The primary objective of Inverse Reinforcement Learning (IRL) is to learn a reward function from demonstrations (Arora & Doshi, 2021; Russell, 1998). In general, conventional IRL methods rely on extensive online trials and errors that can be costly or require a fully known transition model (Abbeel & Ng, 2004; Ratliff et al., 2006; Ziebart et al., 2008; Syed & Schapire, 2007; Boularias et al., 2011; Osa et al., 2018), struggling to scale in many real-world applications. To tackle this problem, this paper studies *offline IRL*, with focus on learning from a previously collected dataset without online interaction with the environment. Offline IRL holds tremendous promise for safety-sensitive applications where manually identifying an appropriate reward is difficult but historical datasets of human demonstrations are readily available (e.g., in healthcare, autonomous driving, robotics, etc.). In particular, since the learned reward function is a succinct representation of an expert's intention, it is useful for policy learning (e.g., in offline Imitation Learning (IL) (Chan & van der Schaar, 2021)) as well as a number of broader applications (e.g., task description (Ng et al., 2000) and transfer learning (Herman et al., 2016)).

This work aims to address a major challenge in offline IRL, namely the *reward extrapolation error*, where the learned reward function may fail to correctly explain the task and misguide the agent in unseen environments. This issue results from the partial coverage of states in the restricted expert demonstrations (i.e., covariate shift) as well as the high-dimensional and expressive function approximation for the reward. It is further exacerbated due to no reinforcement signal for supervision and the intrinsic *reward ambiguity* therein.[1] In fact, similar challenges related to the extrapolation

---

*Part of this work was done when Sheng Yue, Wei Shao, and Sen Lin worked at Arizona State University.
†Corresponding author: `renju@tsinghua.edu.cn`
[1]The reward ambiguity refers to the fact that same behavior can be optimal for many reward functions.

error *in the value function* have been widely observed in offline (forward) RL, e.g., in Kumar et al. (2020); Yu et al. (2020; 2021). Unfortunately, to the best of our knowledge, this challenge remains not well understood in offline IRL, albeit there is some recent progress (Zolna et al., 2020; Garg et al., 2021; Chan & van der Schaar, 2021). Thus motivated, the key question this paper seeks to answer is: "How to devise offline IRL algorithms that can ameliorate the reward extrapolation error effectively?"

We answer this question by introducing a principled offline IRL algorithm, named conservative model-based reward learning (CLARE), leveraging not only (limited) higher-quality expert data but also (potentially abundant) lower-quality diverse data to enhance the coverage of the state-action space for combating covariate shift. CLARE addresses the above-mentioned challenge by appropriately *integrating conservatism* into the learned reward to alleviate the possible misguidance in out-of-distribution states, and improves the reward generalization ability by utilizing a learned dynamics model. More specifically, CLARE iterates between *conservative reward updating* and *safe policy improvement*, and the reward function is updated via improving its values on *weighted* expert and diverse state-actions while in turn cautiously penalizing those generated from model rollouts. As a result, it can encapsulate the expert intention while conservatively evaluating out-of-distribution state-actions, which in turn encourages the policy to visit data-supported states and follow expert behaviors and hence achieves safe policy search.

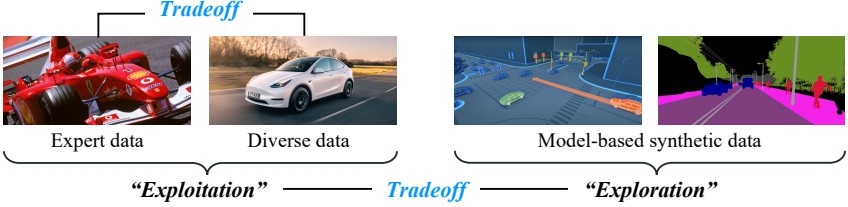

Figure 1: An illustration of the two-tier tradeoffs in CLARE.

Technically, there are highly nontrivial two-tier tradeoffs that CLARE has to delicately calibrate: "balanced exploitation" of the expert and diverse data, and "exploration" of the estimated model.[2] As illustrated in Fig. 1, The first tradeoff arises because CLARE relies on both exploiting expert demonstrations to infer the reward and exploiting diverse data to handle the covariate shift caused by the insufficient state-action coverage of limited demonstration data. At a higher level, CLARE needs to judiciously explore the estimated model to escape the offline data manifold for better generalization. To this end, we first introduce the new *pointwise weight parameters* for offline data points (state-action pairs) to capture the subtle two-tier exploitation-exploration tradeoffs. Then, we rigorously quantify its impact on the performance by providing an upper bound on the return gap between the learned policy and the expert policy. Based on the theoretical quantification, we derive the optimal weight parameters whereby CLARE can strike the balance appropriately to minimize the return gap. Our findings reveal that the reward function obtained by CLARE can effectively capture the expert intention and provably ameliorate the extrapolation error in offline IRL.

Finally, extensive experiments are carred out to compare CLARE with state-of-the-art offline IRL and offline IL algorithms on MuJoCo continuous control tasks. Our results demonstrate that even using small offline datasets, CLARE obtains significant performance gains over existing algorithms in continuous, high-dimensional environments. We also show that the learned reward function can explain the expert behaviors well and is highly instructive for further learning.

## 2 PRELIMINARIES

**Markov decision process (MDP)** can be specified by tuple $M \doteq \langle \mathcal{S}, \mathcal{A}, T, R, \mu, \gamma \rangle$, consisting of state space $\mathcal{S}$, action space $\mathcal{A}$, transition function $T : \mathcal{S} \times \mathcal{A} \to \mathcal{P}(\mathcal{S})$, reward function $R : \mathcal{S} \times \mathcal{A} \to \mathbb{R}$, initial state distribution $\mu : \mathcal{S} \to [0, 1]$, and discount factor $\gamma \in (0, 1)$. A stationary stochastic policy maps states to distributions over actions as $\pi : \mathcal{S} \to \mathcal{P}(\mathcal{A})$. We define the normalized state-action occupancy measure (abbreviated as occupancy measure) of policy $\pi$

---

[2]The exploration in the context of this manuscript refers to enhancing the generalization capability of the algorithm by escaping the offline data manifold via model rollout.

under transition dynamics $T$ as $\rho^\pi(s,a) \doteq (1-\gamma)\sum_{h=0}^\infty \gamma^h \Pr(s_h = s|T,\pi,\mu)\pi(a|s)$. The objective of reinforcement learning (RL) can be expressed as maximizing expected cumulative rewards: $\max_{\pi \in \Pi} J(\pi) \doteq \mathbb{E}_{s,a\sim\rho^\pi}[R(s,a)]$, where $\Pi$ is the set of all stationary stochastic policies that take actions in $\mathcal{A}$ given states in $\mathcal{S}$.[3]

**Maximum entropy IRL (MaxEnt IRL)** aims to learn the reward function from expert demonstrations and reason about the *stochasticity* therein (Ziebart et al., 2008; Ho & Ermon, 2016). Based on demonstrations sampled from expert policy $\pi^E$, the MaxEnt IRL problem is given by

$$\min_{r\in\mathcal{R}}\left(\max_{\pi\in\Pi}\alpha H(\pi) + \mathbb{E}_{s,a\sim\rho^\pi}[r(s,a)]\right) - \mathbb{E}_{s,a\sim\rho^E}[r(s,a)] + \psi(r), \tag{1}$$

with $H(\pi) \doteq -\iint \rho^\pi(s,a)\log\pi(a|s)\,\mathrm{d}s\,\mathrm{d}a$ being the $\gamma$-discounted causal entropy, $\mathcal{R}$ a family of reward functions, $\alpha \geq 0$ the weight parameter, and $\psi : \mathbb{R}^{\mathcal{S}\times\mathcal{A}} \to \mathbb{R} \cup \{\infty\}$ a convex reward regularizer Fu et al. (2018); Qureshi et al. (2018). Problem (1) looks for a reward function assigning higher rewards to the expert policy and lower rewards to other policies, along with the best policy under the learned reward function. Although enjoying strong theoretical justification and achieving great performance in many applications, MaxEnt IRL has to solve a forward RL problem in the inner loop that involves extensive online interactions with the environment.

**Offline IRL** is the setting where the algorithm is neither allowed to interact with the environment nor provided reinforcement signals. It only has access to static dataset $\mathcal{D} = \mathcal{D}_E \cup \mathcal{D}_B$ consisting of expert dataset $\mathcal{D}_E \doteq \{(s_i,a_i,s_i')\}_{i=1}^{D_E}$ and diverse dataset $\mathcal{D}_B \doteq \{(s_i,a_i,s_i')\}_{i=1}^{D_B}$ collected by expert policy $\pi^E$ and behavior policy $\pi^B$, respectively. The goal of offline IRL is to infer a reward function capable of explaining the expert's preferences from the given dataset.

## 3 CLARE: CONSERVATIVE MODEL-BASED REWARD LEARNING

A naive solution for offline IRL is to retrofit MaxEnt IRL to the offline setting via estimating a dynamics model using offline data (e.g., in Tanwani & Billard (2013); Herman et al. (2016)). Unfortunately, it has been reported that this naive paradigm often suffers from unsatisfactory performance in high-dimensional and continuous environments Jarrett et al. (2020). The underlying reasons for this issue include: (1) the dependence on full knowledge of the reward feature function, and (2) the lack of effective mechanisms to tackle the reward extrapolation error caused by covariate shift (as stated in Section 1). Nevertheless, we believe that utilizing a learned dynamics model is beneficial because it is expected to provide broader generalization by learning on additional model-generated synthetic data (Yu et al., 2020; 2021; Lin et al., 2021). With this insight, this work focuses on the model-based offline IRL method that is robust to covariate shift while enjoying the model's generalization ability.

As illustrated in Fig. 1, there are two-tier subtle tradeoffs that need to be carefully balanced between exploiting the offline data and exploring model-based synthetic data. On one hand, the higher-quality expert demonstrations are exploited to infer the intention and abstract the reward function therein, while the lower-quality diverse data is exploited to enrich data support. On the other hand, it is essential to prudently explore the estimated dynamics model to improve the generalization capability while mitigating overfitting errors in inaccurate regions. To this end, we devise conservative model-based reward learning (CLARE) based on MaxEnt IRL, where the new *pointwise weight parameters* are introduced for each offline state-action pair to capture the tradeoffs subtly. We elaborate further in what follows.

As outlined below, CLARE iterates between *(I) conservative reward updating* and *(II) safe policy improvement*, under a dynamics model (denoted by $\widehat{T}$) learned from offline dataset.

*(I) Conservative reward updating.* Given current policy $\pi$, dynamics model $\widehat{T}$, and offline datasets $\mathcal{D}_E$ and $\mathcal{D}_B$, CLARE updates reward funtion $r$ based on the following loss:

$$L(r|\pi) \doteq \underbrace{Z_\beta \mathbb{E}_{s,a\sim\hat{\rho}^\pi}[r(s,a)]}_{\text{penalized on model rollouts}} - \underbrace{\mathbb{E}_{s,a\sim\tilde{\rho}^E}[r(s,a)]}_{\text{increased on expert data}} - \underbrace{\mathbb{E}_{s,a\sim\tilde{\rho}^D}[\beta(s,a)r(s,a)]}_{\text{weighting expert and diverse data}} + \underbrace{Z_\beta\psi(r)}_{\text{regularizer}}, \tag{2}$$

---

[3]For convenience, we omit a constant multiplier, $1/(1-\gamma)$, in the objective for conciseness, i.e., the complete objective function is given by $\max_{\pi\in\Pi}\mathbb{E}_{s,a\sim\rho^\pi}[R(s,a)/(1-\gamma)]$.

where $\tilde{\rho}^D(s,a) \doteq (|\mathcal{D}_E(s,a)| + |\mathcal{D}_B(s,a)|)/(D_E + D_B)$ is the empirical distribution of $(s,a)$ in the union dataset $\mathcal{D} = \mathcal{D}_E \cup \mathcal{D}_B$ and $\tilde{\rho}^E \doteq |\mathcal{D}_E(s,a)|/D_E$ is that for expert dataset $\mathcal{D}_E$; $\hat{\rho}^\pi$ is the occupancy measure when rolling out $\pi$ with dynamics model $\widehat{T}$; and $\psi$ denotes a convex regularizer mentioned above. One key step is to add an additional term weighting the reward of each offline state-action by $\beta(s,a)$, which is a "fine-grained control" for the exploitation of the offline data. For the data deserving more exploitation (e.g., expert behaviors with sufficient data support), we can set a relatively large $\beta(s,a)$; otherwise, we decrease its value. Besides, it can also control the exploration of the model subtly (consider that if we set all $\beta(s,a) = 0$, Eq. (2) reduces to MaxEnt IRL, enabling the agent to explore the model without restrictions). Here, $Z_\beta \doteq 1 + \mathbb{E}_{s',a'\sim\tilde{\rho}^D}[\beta(s',a')]$ is a normalization term. The new ingredients beyond MaxEnt IRL are highlighted in blue.

Observe that in Eq. (2), by decreasing the reward loss, CLARE pushes up the reward on good offline state-action that characterized by larger $\beta(s,a)$, while pushing down the reward on potentially out-of-distribution ones that generated from model rollouts. This is similar to COMBO (Yu et al., 2021) in spirit, a state-of-the-art offline forward RL algorithm, and results in a *conservative reward function*. It can encourage the policy to cautiously exploring the state-actions beyond offline data manifold, thus capable of mitigating the misguidance issue and guiding safe policy search. In Section 4, we will derive a closed-form optimal $\beta(s,a)$ that enables CLARE to achieve a proper exploration-exploitation trade-off by minimizing a return gap from the expert policy.

*(II) Safe policy improvement.* Given updated reward function $r$, the policy is improved by solving

$$\max_{\pi\in\Pi} L(\pi|r) \doteq Z_\beta \mathbb{E}_{s,a\sim\hat{\rho}^\pi}[r(s,a)] + \alpha\widehat{H}(\pi), \tag{3}$$

where $\alpha \geq 0$ is a weight parameter, and $\widehat{H}(\pi) \doteq -\iint \hat{\rho}^\pi(s,a)\log\pi(a|s)\,\mathrm{d}s\,\mathrm{d}a$ is the $\gamma$-discounted causal entropy induced by the policy and learned dynamics model. Due to the embedded expert intention and conservatism in the reward function, the policy is updated safely by carrying out conservative model-based exploration. One can use any well-established MaxEnt RL approach to solve this problem by simulating with model $\widehat{T}$ and reward function $r$. It is worth noting that for Problem (3) in this step, the practical implementation of CLARE works well with a small number of updates in each iteration (see Sections 5 and 6).

## 4 THEORETICAL ANALYSIS OF CLARE

In this section, we focus on answering the following question: "How to set $\beta(s,a)$ for each offline state-action pair to strike the two-tier exploitation-exploration balance appropriately?" To this end, we first quantify the impact of the tradeoffs via bounding the return gap between the learned policy and expert policy. Then, we derive the optimal weight parameters to minimize this gap. All the detailed proofs can be found in Appendix B. Notably, this section works with finite state and action spaces, but our algorithms and experiments run in high-dimensional and continuous environments.

### 4.1 CONVERGENCE ANALYSIS

We first characterize the policy learned by CLARE, in terms of $\beta(s,a)$ and empirical distributions $\tilde{\rho}^E$ and $\tilde{\rho}^D$. Before proceeding, it is easy to see CLARE is iteratively solving the min-max problem:

$$\min_{r\in\mathcal{R}}\max_{\pi\in\Pi} \underbrace{\alpha\widehat{H}(\pi) + Z_\beta\mathbb{E}_{\hat{\rho}^\pi}[r(s,a)] - \mathbb{E}_{\tilde{\rho}^D}[\beta(s,a)r(s,a)] - \mathbb{E}_{\tilde{\rho}^E}[r(s,a)] + Z_\beta\psi(r)}_{\doteq L(\pi,r)}. \tag{4}$$

For dynamics $T$, define the set of occupancy measures satisfying *Bellman flow constraints* as

$$\mathcal{C}_T \doteq \left\{\rho \in \mathbb{R}^{|\mathcal{S}||\mathcal{A}|} : \rho \geq 0 \text{ and } \sum_a \rho(s,a) = \mu(s) + \gamma\sum_{s',a} T(s|s',a)\rho(s',a) \,\forall s\in\mathcal{S}\right\}. \tag{5}$$

We first provide the following results for switching between policies and occupancy measures, which allow us to use $\pi_\rho$ to denote the unique policy for occupancy measure $\rho$.

**Lemma 4.1** (Theorem 2 in Syed et al. (2008)). *If $\rho \in \mathcal{C}_T$, then $\rho$ is the occupancy measure for stationary policy $\pi_\rho(a|s) \doteq \rho(s,a)/\sum_{a'}\rho(s,a')$, and $\pi_\rho$ is the only stationary policy with occupancy measure $\rho$.*

**Lemma 4.2** (Lemma 3.2 in Ho & Ermon (2016)). *Denote* $\bar{H}(\rho) \doteq -\sum_{s,a} \rho(s,a) \log \frac{\rho(s,a)}{\sum_{a'} \rho(s,a')}$. *Then,* $\bar{H}$ *is strictly concave, and for all* $\pi \in \Pi$ *and* $\rho \in \mathcal{C}_T$, $H(\pi) = \bar{H}(\rho^{\pi})$ *and* $\bar{H}(\rho) = H(\pi_\rho)$ *hold true, where* $\pi_\rho(a|s) \doteq \rho(s,a) / \sum_{a'} \rho(s,a')$.

Based on Lemma 4.1 and Lemma 4.2, we have the follow results on the learned policy.

**Theorem 4.1.** *Assume that* $\beta(s,a) \geq -\tilde{\rho}^E(s,a)/\tilde{\rho}^D(s,a)$ *holds for* $(s,a) \in \mathcal{D}$. *For Problem (4), the following relationship holds:*

$$\min_{r \in \mathcal{R}} \max_{\pi \in \Pi} L(\pi, r) = \max_{\hat{\rho} \in \mathcal{C}_{\widehat{T}}} \alpha \bar{H}(\hat{\rho}) - Z_\beta D_\psi \left( \hat{\rho}, \frac{\tilde{\rho}^E + \beta \tilde{\rho}^D}{Z_\beta} \right), \tag{6}$$

*with* $D_\psi(\rho_1, \rho_2) \doteq \psi^*(\rho_2 - \rho_1)$, *where* $\psi^*$ *is the convex conjugate of* $\psi$.

Notably, by selecting appropriate forms of reward regularizers $\psi$, $D_\psi$ can belong to a wide-range of statistical distances. For example, if $\psi(r) = \alpha r^2$, then $D_\psi(\rho_1, \rho_2) = \frac{1}{4\alpha} \chi^2(\rho_1, \rho_2)$; if $\psi$ restricts $r \in [-R^{\max}, R^{\max}]$, then $D_\psi(\rho_1, \rho_2) = 2R^{\max} D_{\text{TV}}(\rho_1, \rho_2)$ (Garg et al., 2021). Theorem 4.1 implies that CLARE implicitly seeks a policy *under* $\widehat{T}$ whose occupancy measure stays close to an interpolation of the empirical distributions of expert dataset $\mathcal{D}_E$ and union offline dataset $\mathcal{D}$. The interpolation reveals that CLARE is trying to trade off the exploration of the model and exploitation of offline data by selecting proper weight parameters $\beta(s,a)$. For example, if $\beta(s,a) = 0$ for all $(s,a) \in \mathcal{D}$, CLARE will completely follow the occupancy measure of the (empirical) expert policy by explore the model freely. In contrast, if $\beta(s,a)$ increases with $\tilde{\rho}^D(s,a)$, the learned policy will look for richer data support.

***Remarks.*** Looking deeper into Eq. (6), the target occupancy measure can be expressed equivalently as $\frac{(1+\beta D_E/D)\tilde{\rho}^E + (\beta D_S/D)\tilde{\rho}^B}{Z_\beta}$, after rearranging terms in the above interpolation. As a result, CLARE also subtly balances the exploitation between the expert and diverse datasets to extract potentially valuable information in the sub-optimal data.

## 4.2 STRIKING THE RIGHT EXPLORATION-EXPLOITATION BALANCE

Next, we show how to set $\beta(s,a)$ properly to achieve the right two-tier balance.

Recall that $J(\pi) \doteq \mathbb{E}_{s,a \sim \rho^\pi}[R(s,a)]$ is the return achieved by policy $\pi$. The next result provides an upper bound on the return gap between $J(\pi)$ and $J(\pi^E)$, which hinges on the intrinsic trade-offs.

**Theorem 4.2.** *Suppose* $|R(s,a)| \leq 1$ *for any* $s \in \mathcal{S}, a \in \mathcal{A}$. *For any stationary policy* $\pi$, *let* $\hat{\rho}^\pi$ *denote the occupancy measure of* $\pi$ *under estimated model* $\widehat{T}$. *We have that*

$$J(\pi^E) - J(\pi) \leq C \cdot \mathbb{E}_{s,a \sim \hat{\rho}^\pi} \left[ D_{\text{TV}} \big( T(\cdot|s,a), \widehat{T}(\cdot|s,a) \big) \right] + 2 \left( D_{\text{TV}}(\hat{\rho}^\pi, \tilde{\rho}^E) + D_{\text{TV}}(\tilde{\rho}^E, \rho^E) \right), \tag{7}$$

*where* $C \doteq \frac{2\gamma}{1-\gamma}$, *and* $\rho^E$ *is the occupancy measure of expert policy* $\pi^E$ *under true dynamics* $T$.

***Remarks.*** Theorem 4.2 indicates that a good policy learned from the estimated model not only follows the expert behaviors but also keeps in the "safe region" of the learned model, i.e., visiting the state-actions with less model estimation inaccuracy. Under the *concentration* assumption, the following holds with probability greater than $1 - \delta$:

$$J(\pi^E) - J(\pi) \leq \underbrace{\mathbb{E}_{s,a \sim \hat{\rho}^\pi} \left[ \frac{CC_\delta}{\sqrt{|\mathcal{D}_E(s,a)| + |\mathcal{D}_B(s,a)|}} \right]}_{(a)} + 2 \underbrace{D_{\text{TV}}(\hat{\rho}^\pi, \tilde{\rho}^E)}_{(b)} + 2 \underbrace{D_{\text{TV}}(\tilde{\rho}^E, \rho^E)}_{(c)},$$

where $\mathcal{D}(s,a) \doteq \{(s',a') \in \mathcal{D} : s' = s, a' = a\}$. It aligns well with the aforementioned exploration-exploitation balance: 1) Term (a) captures the exploitation of offline data support; 2) Term (b) captures the exploitation of expert data and the exploration of the model (recall that $\hat{\rho}^\pi$ is the occupancy measure of rolling out $\pi$ with $\widehat{T}$); and 3) Term (c) captures the distributional shift in offline learning. Importantly, the result in Theorem 4.2 connects the true return of a policy with its occupancy measure on the learned model. This gives us a criteria to evaluate the performance of a policy from offline. Define $c(s,a) \doteq C \cdot D_{\text{TV}}(T(\cdot|s,a), \widehat{T}(\cdot|s,a))$ and $c^{\min} \doteq \min_{s,a} c(s,a)$. Subsequently, we derive the policy that minimizes the RHS of Eq. (7).

---

**Algorithm 1:** Conservative model-based reward learning (CLARE)

---

**Input:** expert data $\mathcal{D}_E$, diverse data $\mathcal{D}_B$, bar $u$, learning rate $\eta$, policy regularizer weight $\lambda$

Learn dynamics model $\widehat{T}$ represented by an ensemble of neural networks using all offline data;

Set weight $\beta(s,a)$ for each offline state-action tuple $(s,a) \in \mathcal{D}_E \cup \mathcal{D}_B$ by Eq. (11);

Initialize the policy $\pi_\theta$ and reward function $r_\phi$ parameterized by $\theta$ and $\phi$ respectively;

**while** *not done* **do**

    (Safe policy improvement) Run a MaxEnt RL algorithm for some steps with model $\widehat{T}$ and current reward function $r_\phi$ to update policy $\pi_\theta$, based on $L(\pi_\theta|r_\phi) - \lambda D_{\mathrm{KL}}(\pi^b \| \pi_\theta)$;

    (Conservative reward updating) Update $r_\phi$ by $\phi \leftarrow \phi - \eta \nabla_\phi L(r_\phi|\pi_\theta)$ for a few steps;

**end**

---

**Theorem 4.3.** *Under the same conditions as in Theorem 4.2, the optimal occupancy measure minimizing the upper bound of Eq. (7) is given as follows:*

$$\hat{\rho}^*(s,a) = \begin{cases} \tilde{\rho}^E(s,a) + \Delta_\rho, & \text{if } c(s,a) \leq c^{\min}, \\ 0, & \text{if } c(s,a) > c^{\min} + 2, \\ \tilde{\rho}^E(s,a), & \text{otherwise}. \end{cases} \tag{8}$$

*where $\Delta_\rho \doteq \frac{\sum_{s',a'} \mathbf{1}[c(s',a') - c^{\min} > 2] \cdot \tilde{\rho}^E(s',a')}{|\mathcal{N}_{\min}|}$ and $\mathcal{N}_{\min} \doteq \{(s,a) \in \mathcal{D} : c(s,a) \leq c^{\min}\}$.*

As shown in Theorem 4.3, the "optimal" policy leaned on model $\widehat{T}$ conservatively explores the model by avoiding the visit of risky state-actions. Meantime, it cleverly exploits the accurate region, such that it does not deviate large from the expert. Now, we are ready to derive the optimal values of the weight parameters.

**Corollary 4.1.** *Suppose that when $\tilde{\rho}^D(s,a) = 0$, $c(s,a) > c^{\min}$ holds for each $(s,a) \in \mathcal{S} \times \mathcal{A}$. Under the same condition as in Theorem 4.3, if $\beta(s,a)$ are set as*

$$\beta^*(s,a) = \begin{cases} \frac{\Delta_\rho}{\tilde{\rho}^D(s,a)}, & \text{if } c(s,a) \leq c^{\min} \text{ and } \tilde{\rho}^D(s,a) > 0, \\ -\frac{\tilde{\rho}^E(s,a)}{\tilde{\rho}^D(s,a)}, & \text{if } c(s,a) > c^{\min} + 2 \text{ and } \tilde{\rho}^D(s,a) > 0, \\ 0, & \text{otherwise}, \end{cases} \tag{9}$$

*then it follows that*

$$\min_{r \in \mathcal{R}} \max_{\pi \in \Pi} L(\pi, r) = \max_\pi \alpha \bar{H}(\hat{\rho}^\pi) - Z_\beta D_\psi(\hat{\rho}^\pi, \hat{\rho}^*). \tag{10}$$

Corollary 4.1 provides the value of $\beta(s,a)$ for each $(s,a) \in \mathcal{D}$ such that the learned reward function can guide the policy to minimize the return gap in Eq. (7). It indicates that the right exploitation-exploration trade-off can be provably balanced via setting the weight parameters properly. In particular, $\beta^*$ assigns positive weight to the offline state-action with accurate model estimation and negative weight to that with large model error. It enables CLARE to learn a conservative reward function that pessimistically evaluates the our-of-distribution states and actions, capable of ameliorating the extrapolation error in unseen environments. However, the optimal weights require the model error, $c(s,a)$, which is typically hard to obtain (especially in high-dimensional and continuous spaces). Section 5 will solve this problem by extending this result with the aid of the model ensembles and uncertainty quantification techniques.

## 5 PRACTICAL IMPLEMENTATION

**Learning dynamics models.** Following the state-of-the-art model-based methods (Yu et al., 2020; 2021), we model the transition dynamics by an ensemble of neural networks, each of which outputs a Gaussian distribution over next states, i.e., $\{\widehat{T}_i(s'|s,a) = \mathcal{N}(\mu_i(s,a), \Sigma_i(s,a))\}_{i=1}^N$.

**Weights in continuous environments.** The ideas of achieving CLARE in continuous environments are 1) to approximately see the offline data as sampled from a large discrete space, and 2)

to use an uncertainty quantification technique for quantifying the model error. Specifically, because state-action pairs are basically different from each other in this setting, we let $\tilde{\rho}^D(s, a) = 1/D$ and $\tilde{\rho}^E(s, a) = 1/D_E$, and employ the uncertainty estimator, $c(s, a) = \max_{i \in [N]} \|\Sigma_i(s, a)\|_F$, proposed in Yu et al. (2020) for model error evaluation. Guided by the analytical results in Corollary 4.1, we compute the weights for each $(s, a) \in \mathcal{D}$ via slight relaxation as follows:

$$\beta(s, a) = \begin{cases} \frac{N'' D}{N' D_E}, & \text{if } c(s, a) \leq u, \\ -\frac{D}{D_E} \cdot \mathbf{1}[(s, a) \in \mathcal{D}_E], & \text{if } c(s, a) > u, \\ 0, & \text{otherwise}, \end{cases} \quad (11)$$

where $N' \doteq \sum_{(s,a) \in \mathcal{D}} \mathbf{1}[c(s, a) \leq u]$ and $N'' \doteq \sum_{(s,a) \in \mathcal{D}_E} \mathbf{1}[c(s, a) > u]$. Here, coefficient $u$ is a user-chosen hyper-parameter for controlling the conservatism level of CLARE. If one wants the learned policy to be trained more conservatively on offline data support, $u$ should be small; otherwise, $u$ can be chose to be large for better exploration.

**Reward and policy regularizers.** In the experiments, we use $\psi(r) = r^2$ as the reward regularizer. Additionally, when updating the policy, we use a KL divergence as a regularizer with empirical behavior policy $\pi^b$ induced by a subset of the offline dataset, $\mathcal{D}' \subset \mathcal{D}$, as follows:

$$D_{\text{KL}}(\pi^b \| \pi) \doteq \mathbb{E}_{s \in \mathcal{D}'} \Big[ \mathbb{E}_{a \sim \pi^b(\cdot|s)} \big[ \log \pi^b(a|s) \big] - \mathbb{E}_{a \sim \pi^b(\cdot|s)} \big[ \log \pi(a|s) \big] \Big],$$

where $\pi^b(a|s) = \frac{\sum_{(s',a') \in \mathcal{D}'} \mathbf{1}[s'=s, a'=a]}{\sum_{(s',a') \in \mathcal{D}'} \mathbf{1}[s'=s]}$ if $(s, a) \in \mathcal{D}'$, and $\pi^b(a|s) = 0$ otherwise. It can be implemented by adding $-\mathbb{E}_{s,a \sim \mathcal{D}'}[\log \pi(a|s)]$ to the actor loss. The intuition is to encourage the actor to perform in support of the real data for accelerating safe policy improvement. While this regularization lacks theoretical guarantees, we empirically find that it can indeed speed up the training.

**Practical algorithm design.** The pseudocode of CLARE is depicted in Algorithm 1. The policy improvement phase can be implemented by the standard implementation of SAC (Haarnoja et al., 2018) with a change of the additional policy regularizer. We elaborate more details in the Appendix A.

## 6 EXPERIMENTS

Next, we use experimental studies to evaluate CLARE and answer the following key questions: (1) How does CLARE perform on the standard offline RL benchmarks in comparison to existing state-of-the-art algorithms? (2) How does CLARE perform given different dataset sizes? (3) How does the "conservatism level", $u$, affect the performance? (4) How fast does CLARE converge? (5) Can the learned reward function effectively explain the expert intention?

To answer these questions, we compare CLARE with the following existing offline IRL methods on the D4RL benchmark (Fu et al., 2020): 1) IQ-LEARN (Garg et al., 2021), a state-of-the-art model-free offline IRL algorithm; 2) AVRIL (Chan & van der Schaar, 2021), another recent model-free offline IRL method; 3) EDM (Jarrett et al., 2020), a state-of-the-art offline IL approach; and 4) Behavior Cloning (BC). To demonstrate the poor performance of the naive approach using a simple combination of IRL with model-based offline forward RL (MORL) method, we also consider a baseline algorithm, namely MOMAX, by directly using COMBO (Yu et al., 2021) in the inner loop of MaxEnt IRL. We present the results on continuous control tasks (including Half-Cheetah, Walker2d, Hopper, and Ant) consisting of three data qualities (random, medium, and expert). Experimental set-up and hyperparameters are described in detailed in Appendix A.

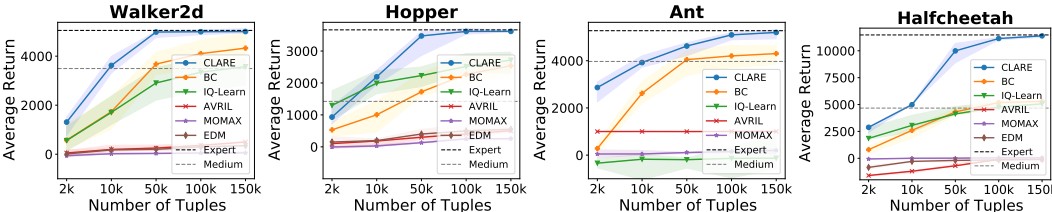

Figure 2: CLARE against other algorithms on all tasks over different dataset sizes consisting of expert and medium data equally.

Table 1: *Results on D4RL datasets.* For each task, the experiments are carried out with three different data combinations: 1) 10k expert tuples, 2) 5k expert and 5k medium tuples, and 3) 5k expert and 5k random tuples. The data scores below for 1), 2), and 3) correspond to expert, medium, and random data, respectively. We tune IQ-LEARN, EDM, and AVRIL based on their publicly available source code. Results are averaged over 7 random seeds. The highest score across all algorithms is bold.

| Dataset type | Environment | Data score | CLARE | BC | IQ-LEARN | EDM | AVRIL | MOMAX |
|---|---|---|---|---|---|---|---|---|
| *Exp. & Rand.* | Walker2d | 1.9 | **2873.8** | 17.8 | 256.9 | 165.5 | 100.9 | -525.4 |
| | Hopper | 18.4 | **1891.5** | 110.2 | 523.6 | 178.8 | 178.3 | 0.7 |
| | Ant | -64.4 | **1960.0** | -427.6 | -247.2 | -3000.9 | 1000.1 | 113.8 |
| | Half-Cheetah | -505.1 | **1113.7** | -86.7 | 123.9 | -346.7 | -1093.5 | -11.0 |
| *Exp. & Med.* | Walker2d | 3496.3 | **3613.4** | 1674.2 | 1676.8 | 175.7 | 184.0 | 19.6 |
| | Hopper | 1422.7 | **2135.0** | 947.0 | 2049.8 | 194.4 | 183.7 | 27.6 |
| | Ant | 3969.0 | **3879.4** | 2146.0 | 222.2 | -3001.5 | 1001.0 | -33.2 |
| | Half-Cheetah | 4667.8 | **4888.6** | 2375.0 | 2957.7 | -298.3 | -1195.6 | -0.2 |
| *Exp.* | Walker2d | 5010.4 | **4990.5** | 1665.7 | 2445.4 | 189.7 | 194.1 | 23.2 |
| | Hopper | 3603.2 | 2604.5 | 1436.1 | **2854.4** | 192.5 | 183.9 | 34.5 |
| | Ant | 5172.8 | **3940.3** | 1797.9 | 375.4 | -3000.6 | 1000.2 | 48.1 |
| | Half-Cheetah | 10748.7 | **4975.1** | 242.4 | 3750.5 | -299.5 | -619.0 | -0.4 |

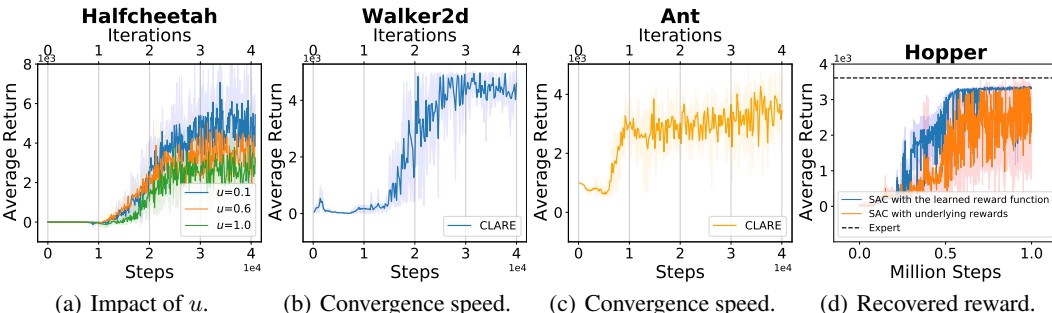

(a) Impact of $u$.   (b) Convergence speed.   (c) Convergence speed.   (d) Recovered reward.

Figure 3: *Performance of CLARE.* 1) *Impact of $u$:* Figure 3(a) shows the impact of user-chosen parameter $u$ on the performance using 10k expert tuples. 2) *Convergence speed:* Figures 3(c) and 3(b) show the convergence of CLARE using 10k expert and 10k medium tuples. In each iteration, CLARE carries out policy improvement by total 10k gradient updates (total 500 epochs with 20 gradient steps per epoch) for the actor and critic networks using SAC. 3) *Recovered reward:* Figure 3(d) shows the result of training SAC via replacing the underlying reward by the one learned from CLARE.

**Results on MuJoCo control.** To answer the first question and validate the effectiveness of the learned reward, we evaluate CLARE on different tasks using limited state-action tuples sampled from D4RL datasets. The ranges of standard deviations of the results in *Exp. & Rand.*, *Exp. & Med.* and *Exp.* are 156.4-280.5, 15.7-127.8 and 42.4-89.5, respectively. As shown in Table 6, CLARE yields the best performance by a significant margin on almost all datasets, especially with low-quality data thereof. It demonstrates that the reward function learned by CLARE can effectively guide offline policy search while exploiting the useful knowledge in the diverse data.

**Results under different dataset sizes.** To answer the second question, we vary the total numbers of state-action tuples from 2k to 100k and present the results on different tasks in Figure 2. CLARE reaches expert performance on each task with sufficient data. Albeit with very limited data, CLARE also achieves strong performance over existing algorithms, revealing its great sample efficiency.

**Results under different $u$.** To answer the third question, we normalize the uncertainty measure to $[0, 1]$ and vary $u$ from 0.1 to 1.0. Due to Eq. (11), a smaller $u$ corresponds to a more conservative CLARE. As illustrated in Figure 3(a), the performance becomes better with the decrease of $u$ value. It validates the importance of the embedded conservatism in alleviating the extrapolation error. We empirically find that the performance with respect to $u$ varies in different tasks. Thus, we treat it as a hyper-parameter to tune In practice.

**Convergence speed.** To answer the fourth question, we present the results on the convergence speed of CLARE in Figure 3(b), revealing its great learning efficiency. It showcases that CLARE converges in 5 iterations with totally less than 50k gradient steps.

**Recovered reward function.** To answer the last question, we evaluate the learned reward function by transferring it to the real environment. As demonstrated in Figure 3(c), the reward function is highly instructive for online learning. It implies that it can effectively reduce the reward extrapolation error and represent the task preferences well. Surprisingly, compared to the true reward function, the policy trained via the learned one performs more stably. The reason is that the learned one incorporates conservatism and thus is capable of penalizing risks and guide safe policy search.

# 7 RELATED WORK

**Offline IRL.** To side-step the expensive online environmental interactions in classic IRL, offline IRL aims to infer a reward function and recover the expert policy only from a static dataset with no access to the environment. Klein et al. (2011) extend the classic apprenticeship learning (i.e., Abbeel & Ng (2004)) to batch and off-policy cases by introducing a temporal difference method, namely LSTD-$\mu$, to compute the feature expectations therein. Klein et al. (2012) further introduce a linearly parameterized score function-based multi-class classification algorithm to output reward function based on an estimate of expert feature expectation. Herman et al. (2016) present a gradient-based solution that simultaneously estimates the feature weights and parameters of the transition model by taking into account the bias of the demonstrations. Lee et al. (2019) propose Deep Successor Feature Networks (DSFN) that estimates feature expectations in an off-policy setting. However, the assumption of full knowledge of the reward feature functions in Klein et al. (2011); Herman et al. (2016); Lee et al. (2019); Jain et al. (2019); Pirotta & Restelli (2016); Ramponi et al. (2020) is often unrealistic, because the choice of features is problem-dependent and can become a very hard task for complex problems (Arora & Doshi, 2021; Piot et al., 2014). To address this problem, Piot et al. (2014) propose a non-parametric algorithm, called RCAL, using boosting method to minimize directly the criterion without the step of choosing features. Konyushkova et al. (2020) propose two semi-supervised learning algorithms that learn a reward function from limited human reward annotations. Zolna et al. (2020) further propose ORIL that can learn from both expert demonstrations and a large unlabeled set of experiences without human annotations. Chan & van der Schaar (2021) use a variational method to jointly learn an approximate posterior distribution over the reward and policy. Garg et al. (2021) propose an off-policy IRL approach, namely IQ-Learn, implicitly representing both reward and policy via a learned soft Q-function. Nevertheless, these methods primarily concentrate on offline policy learning with learning reward function being an intermediate step. Due to the intrinsic covariate shift, these methods may suffer from severe reward extrapolation error, leading to misguidance in unseen environments and low learning efficiency.

**Offline IL.** Akin to offline IRL, offline imitation learning (offline IL) deals with training an agent to directly mimic the actions of a demonstrator in an entirely offline fashion. Behavioral cloning (BC (Ross & Bagnell, 2010)) is indeed an intrinsically offline solution, but it fails to exploit precious dynamics information. To tackle this issue, several recent works propose dynamics-aware offline IL approaches, e.g., Kostrikov et al. (2019); Jarrett et al. (2020); Chang et al. (2021); Swamy et al. (2021). In contrast to directly mimicking the expert as done in offline IL, offline IRL explicitly learns the expert's reward function from offline datasets, which can take into account the temporal structure and inform what the expert wishes to achieve, rather than simply what they are reacting to. It enables agents to understand and generalize these "intentions" when encountering similar environments and therefore makes offline IRL more robust (Lee et al., 2019). In addition, the learned reward function can succinctly explain the expert's objective, which is also useful in a number of broader applications (e.g., task description Ng et al. (2000) and transfer learning Herman et al. (2016)).

# 8 CONCLUSION

This paper introduces a new offline IRL algorithm (namely CLARE) to approaching the reward extrapolation error (caused by covariate shift) via incorporating conservatism into a learned reward function and utilizing an estimated dynamics model. Our theoretical analysis characterizes the impact of covariate shift by quantifying a subtle two-tier exploitation-exploration tradeoffs, and we show that CLARE can provably alleviate the reward extrapolation error by striking the right tradeoffs therein. Extensive experiments corroborate that CLARE outperforms existing methods in continuous, high-dimensional environments by a significant margin, and the learned reward function represents the task preferences well.

ACKNOWLEDGMENTS

This research was supported in part by the National Natural Science Foundation of China under Grant No. 62122095, 62072472, and U19A2067, by NSF Grants CNS-2203239, CNS-2203412, and RINGS-2148253, and by a grant from the Guoqiang Institute, Tsinghua University.

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

# A EXPERIMENTAL DETAILS

In this section, we present necessary experimental details for reproducibility.

## A.1 PRACTICAL IMPLEMENTATION DETAILS

In the experiment, our implementation is built upon the open source framework of offline RL algorithms, provided at: https://github.com/polixir/OfflineRL, including data sampling, policy testing, dynamics model structure, etc. The implementation of SAC in the policy improvement uses the open source code available at: https://github.com/pranz24/pytorch-soft-actor-critic (under the MIT License). Additionally, the expert and diverse state-action pairs are sampled at random from the D4RL dataset provided at: https://github.com/rail-berkeley/d4rl (under the Apache License 2.0).

**Model learning.** Following the same line as in Yu et al. (2020; 2021), we model the transition dynamics by an ensemble of probabilistic neural networks, each of which takes the current state and action as input and outputs a Gaussian distribution over next states, i.e., $\{\widehat{T}_i(s'|s,a) = \mathcal{N}(\mu_i(s,a), \Sigma_i(s,a))\}_{i=1}^N$. Using offline state-action pairs, 7 models are trained independently via maximum likelihood, each of which is represented as by a 4-layer feedforward neural network with 256 hidden units. The best 5 models are picked based on the validation prediction error on a held-out set. During model rollouts, one model will be selected randomly from the ensemble.

**Policy improvement.** We represent both critic and actor as a 2-layer feedforward neural network with 256 hidden units and Swish activation functions. In each iteration, we update the critic and actor networks using SAC (Haarnoja et al., 2018) for 500 epochs (each has 20 gradient updates). As described in Section 5, we use a KL divergence with the behavior policy to accelerate inner-loop policy search. It is implemented by adding $-\mathbb{E}_{s,a\sim\mathcal{D}'}[\log\pi(a|s)]$ ($\mathcal{D}' \in \mathcal{D}$) to the actor loss. An instantiation of the policy improvement can be found in Algorithm 2.

---

**Algorithm 2:** Safe policy improvement

**Input:** offline dataset $\mathcal{D}$, policy regularizer weight $\lambda$, rollout horizon $H$, rollout batchsize $B$, the number of epochs $E$, dynamics ensemble $\{\widehat{T}_i\}_{i=1}^N$, reward function $r_\phi$, policy $\pi_\theta$
Initialize model buffer $\mathcal{D}_{\text{model}} \leftarrow \varnothing$;
**for** *epoch* $= 1$ **to** $E$ **do**
  **for** $b = 1$ **to** $B$ **in parallel do**
    Sample state $s_1$ from $\mathcal{D}$ as the initial state of the rollout;
    **for** $h = 1$ **to** $H$ **do**
      Sample action $a_h \sim \pi_\theta(\cdot|s_h)$;
      Randomly pick dynamics $\widehat{T}$ from $\{\widehat{T}_i\}_{i=1}^N$ and sample $s_{h+1} \sim \widehat{T}(\cdot|s_h, a_h)$;
      Compute $r_h \leftarrow r_\phi(s_h, a_h)$;
      Add sample $(s_h, a_h, r_h, s_{h+1})$ to $\mathcal{D}_{\text{model}}$;
    **end**
  **end**
  Sample batches from $\mathcal{D}_{\text{model}}$ and use SAC to update policy $\pi_\theta$ with $-\mathbb{E}_{s,a\sim\mathcal{D}'}[\log\pi_\theta(a|s)]$
  ($\mathcal{D}' \in \mathcal{D}$) added on the policy loss;
**end**

---

**Reward updating.** We represent the reward function as a 4-layer feedforward neural network with 256 hidden units and Swish activate functions. In each iteration, the reward function is updated by 5 gradient steps with stepsize $5 \times 10^{-5}$, based on the following practical reward loss:

$$L(r_\phi) \doteq Z_\beta \mathbb{E}_{\mathcal{D}_{\text{replay}}}\left[r_\phi(s,a)\right] + Z_\beta \mathbb{E}_{s,a\sim\mathcal{D}\cup\mathcal{D}_{\text{replay}}}\left[r_\phi(s,a)^2\right]$$
$$- \mathbb{E}_{s,a\sim\mathcal{D}_E}\left[r_\phi(s,a)\right] - \mathbb{E}_{s,a\sim\mathcal{D}}\left[\beta(s,a)r_\phi(s,a)\right]. \quad (12)$$

We use replay buffer $\mathcal{D}_{\text{replay}}$ across iterations to save the simulated data for training stability. An instantiation of reward updating is shown in Algorithm 3.

**Practical algorithm.** Based on Algorithms 2 and 3, a detailed CLARE algorithm is outlined in Algorithm 4.

---

**Algorithm 3:** Conservative reward updating

---

**Input:** expert data $\mathcal{D}_E$, diverse data $\mathcal{D}_B$, replay buffer $\mathcal{D}_{\text{replay}}$, model buffer $\mathcal{D}_{\text{model}}$, reward function $r_\phi$, learning rate $\eta$, the number of steps $T$

Update replay buffer $\mathcal{D}_{\text{replay}} \leftarrow \mathcal{D}_{\text{replay}} \cup \mathcal{D}_{\text{model}}$;

**for** $t = 1$ **to** $T$ **do**
   | Update the parameters of reward function $r_\phi$ by $\phi \leftarrow \phi - \eta \nabla L(r_\phi)$;
**end**

---

**Algorithm 4:** Conservative model-based reward learning (CLARE)

---

**Input:** expert data $\mathcal{D}_E$, diverse data $\mathcal{D}_B$, bar $u$, learning rate $\eta$, policy regularizer weight $\lambda$

Learn dynamics model $\widehat{T}$ represented by an ensemble of neural networks using all offline data;

Set weight $\beta(s, a)$ for each offline state-action tuple $(s, a) \in \mathcal{D}_E \cup \mathcal{D}_B$ by Eq. (11);

Initialize the policy $\pi_\theta$ and reward function $r_\phi$ parameterized by $\theta$ and $\phi$ respectively;

Initialize replay buffer $\mathcal{D}_{\text{replay}} \leftarrow \varnothing$;

**while** *not done* **do**
   | (Safe policy improvement) Run Algorithm 2 to update policy $\pi_\theta$ and get model buffer
   |   $\mathcal{D}_{\text{model}}$;
   | (Conservative reward updating) Run Algorithm 3 to update reward function $r_\phi$;
**end**

---

## A.2 HYPERPARAMETERS

We summarize the hyperparameters used in the evaluation as follows.

**Conservatism level $u$.** For all tasks, we normalize the uncertainty measure to $[0, 1]$ and test $u$ from set $\{0.4, 0.6, 0.8\}$. The result is shown in Table 3. In each experiment, we select the $u$ value that achieves the maximum corresponding score.

**Learning rates.** For all experiments, the reward learning rate is $\eta = 5 \times 10^{-5}$. Our empirical studies indicate that a relatively small reward learning rate leads to more stable training. Additionally, the learning rates for actor and critic are both $3 \times 10^{-4}$, and that for dynamics model is $10^{-3}$.

**Policy regularization.** For all experiments, the policy regularization weight is $\lambda = 0.25$.

The additional hyperparameters are listed in Table A.2.

Table 2: *Hyperparameters for CLARE.* Instead of $u$, the hyperparameters used in the evaluation are identical across different tasks (Half-Cheetah, Walker2d, Hopper, and Ant).

| Hyperparameter | Value |
|---|---|
| Reward learning rate ($\eta$) | $5 \times 10^{-5}$ |
| Rollout batchsize ($B$) | 5000 |
| Rollout horizon ($H$) | 5 |
| Policy regularization weight ($\lambda$) | 0.25 |
| Discount factor ($\gamma$) | 0.99 |
| # steps per reward updating ($T$) | 5 |
| # epochs ($E$) | 500 |
| # steps per epoch | 20 |
| Actor learning rate | $3 \times 10^{-4}$ |
| Critic learning rate | $3 \times 10^{-4}$ |

## A.3 MORE EXPERIMENTAL RESULTS

We further evaluate CLARE by answering the following two questions: 1) Can CLARE exploit the useful information from diverse datasets? 2) How does CLARE perform compared to the simple

Table 3: *Performance under different $u$ values.* We tune $u$ from set $\{0.4, 0.6, 0.8\}$. For each MuJoCo task, the experiments are carried out with three data combinations: 1) 10k expert state-action tuples, 2) 5k expert and 5k medium state-action tuples, and 3) 5k expert and 5k random state-action tuples. The highest score across different $u$ is bold.

| Dataset type | Environment | $u = 0.4$ | $u = 0.6$ | $u = 0.8$ |
|---|---|---|---|---|
| *Exp. & Rand.* | Walker2d | 2896.93 | **2989.79** | 1083.17 |
| | Hopper | 1187.22 | **1841.15** | 1508.09 |
| | Ant | **2047.98** | 1496.09 | 1337.01 |
| | Half-Cheetah | 453.03 | **1118.58** | 849.42 |
| *Exp. & Med.* | Walker2d | 3334.55 | **3680.78** | 3275.21 |
| | Hopper | 1722.44 | **2107.90** | 1963.59 |
| | Ant | 3568.49 | **3805.64** | 2635.30 |
| | Half-Cheetah | **4955.23** | 4349.88 | 4000.17 |
| *Exp.* | Walker2d | 4674.52 | **4958.04** | 4742.20 |
| | Hopper | 1954.04 | **2605.82** | 2328.22 |
| | Ant | 2747.10 | **3925.90** | 3330.48 |
| | Half-Cheetah | **5050.05** | 4942.20 | 4542.45 |

combination of MORL and (online) IRL methods? 3) What is the impact of reward weighting? 4) What is the impact of expert sample sizes?

**Exploitation on diverse data.** Table 4 shows the results under different data combinations. By using additional medium data, the performance can be improved over that only using 5k expert tuples. The underlying rationale is: 1) The diverse datasets contain some good state-actions; 2) the diverse data support enables CLARE to safely generalize to the states beyond expert data manifold.

Table 4: *Impact of diverse data.* For each MuJoCo task, the experiments are carried out with three data combinations: 1) 10k expert state-action tuples, 2) 5k expert state-action tuples, 3) 5k expert and 5k medium state-action tuples, and 4) 5k expert and 5k random state-action tuples.

| Task | *Exp. (5k) & Rand. (5k)* | *Exp. (5k) & Med. (5k)* | *Exp. (5k)* | *Exp. (10k)* |
|---|---|---|---|---|
| Walker2d | 2973.88 | 3613.49 | 2858.29 | 4990.57 |
| Hopper | 1891.55 | 2135.07 | 1885.76 | 2604.59 |
| Ant | 1960.05 | 3879.48 | 1978.08 | 3940.30 |
| Half-Cheetah | 1113.75 | 4888.64 | 1714.30 | 4975.17 |

**Expert sample sizes.** Table 5 shows the average returns (over 5 random seeds) under different expert sample sizes with the fixed number of medium data (50k). It can corroborate our analytical results that with a relatively sufficient data coverage of the empirical expert behaviors, the performance is dominated by the expert sample size (combining Theorem 4.2, Theorem 4.3 and Corollary 4.1).

Table 5: *Results under different expert sample sizes.*

| Dataset | *2k* | *5k* | *10k* | *20k* | *50k* | *100k* |
|---|---|---|---|---|---|---|
| Half-Cheetah | 4753.9 | 4978.2 | 5206.5 | 7865.5 | 10930.1 | 11121.9 |
| Hopper | 1989.9 | 2273.4 | 2507.8 | 2991.8 | 3571.1 | 3566.4 |
| Walker | 3439.9 | 3632.2 | 4753.3 | 4982.4 | 4977.8 | 4991.3 |
| Ant | 3375.4 | 3866.5 | 3968.9 | 4385.7 | 4797.5 | 4910.6 |

**Comparison to MOMAX.** To demonstrate the poor performance of the naive approach using a simple combination of IRL with model-based offline forward RL (MORL) method, we design a baseline directly using a state-of-the-art MORL method, COMBO (Yu et al., 2021), in the inner loop of MaxEnt IRL (Eq. (1)), called MOMAX. As shown in Figure 4, MOMAX does not work well in

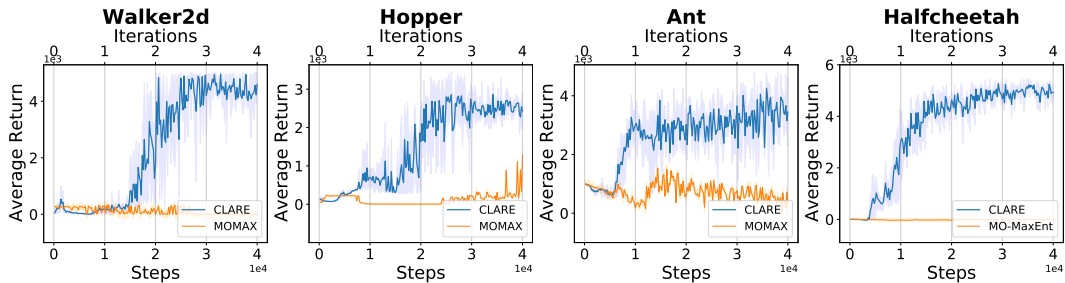

Figure 4: *Comparison to MOMAX.* Each experiment uses 10k expert and 10k medium state-actions.

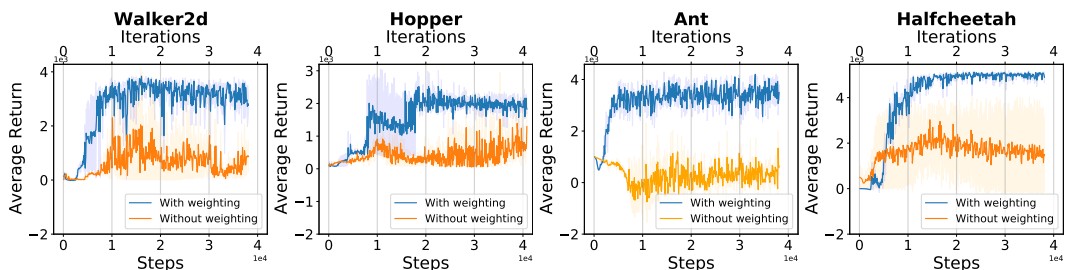

Figure 5: *Ablation study of reward weighting.* Each experiment uses 5k expert and 50k medium state-action samples from the D4RL benchmark.

these continuous control tasks. It reveals the challenges of repurposing the online IRL methods in the offline IRL setting.

**Ablation study of reward weighting.** Fig. 5 shows the impact of reward weighting on performance. CLARE basically reduces to MaxEnt IRL with no reward weighting and thus can not deal with the extrapolation error effectively in offline learning. Fig. 5 also demonstrates that the conservative reward function can stabilize the training.

### A.4 COMPUTATIONAL COMPLEXITY

We implement the code in PyTorch 1.11.0 on a server with a 32-Cores AMD Ryzen Threadripper PRO 3975WX and a Intel GeForch RTX 3090 Ti. For all tasks, CLARE converges in one hour (around 5-10 iterations with total 50k-100k gradient steps).

## B PROOFS

In this section, we provide detailed proofs of main results in Section 4.

### B.1 PROOF OF THEOREM 4.1

This proof is built on that for Ho & Ermon (2016, Proposition 3.1).

First, it follows from Eq. (4) that

$$
\begin{aligned}
L(\pi, r) &= \alpha \widehat{H}(\pi) + Z_\beta \mathbb{E}_{\hat{\rho}^\pi}\left[r(s,a)\right] - \mathbb{E}_{\tilde{\rho}^D}\left[\beta(s,a)r(s,a)\right] - \mathbb{E}_{\tilde{\rho}^E}\left[r(s,a)\right] + Z_\beta \psi(r) \\
&= \alpha \widehat{H}(\pi) + \sum_{s,a}\left(Z_\beta \hat{\rho}^\pi(s,a) - \tilde{\rho}^D(s,a)\beta(s,a) - \tilde{\rho}^E(s,a)\right) r(s,a) + Z_\beta \psi(r) \\
&= \alpha \widehat{H}(\pi) + Z_\beta \sum_{s,a}\left(\hat{\rho}^\pi(s,a) - \frac{\tilde{\rho}^D(s,a)\beta(s,a) + \tilde{\rho}^E(s,a)}{Z_\beta}\right) r(s,a) + Z_\beta \psi(r)
\end{aligned}
$$

$$= \alpha \widehat{H}(\pi) + Z_\beta \left( \mathbb{E}_{\hat{\rho}^\pi}\big[r(s,a)\big] - \mathbb{E}_{\tilde{\rho}^I}\big[r(s,a)\big] + \psi(r) \right).$$

$$\text{(denoting } \tilde{\rho}^I(s,a) \doteq \tfrac{\tilde{\rho}^D(s,a)\beta(s,a)+\tilde{\rho}^E(s,a)}{Z_\beta})$$

where the last equality holds due to

$$
\begin{aligned}
Z_\beta &= 1 + \mathbb{E}_{s,a \sim \tilde{\rho}^D}[\beta(s,a)] \\
&= \sum_{s,a} \tilde{\rho}^E(s,a) + \tilde{\rho}^D(s,a)\beta(s,a) \\
&\geq 0. \qquad\qquad \text{(from } \beta(s,a) \geq -\tilde{\rho}^E(s,a)/\tilde{\rho}^D(s,a) \text{ for } (s,a) \in \mathcal{D})
\end{aligned}
$$

Thanks to Lemma 4.1, there exists a one-to-one correspondence between $\Pi$ and $\mathcal{C}_{\widehat{T}}$. Thus, we can rewrite

$$\min_{r\in\mathcal{R}} \max_{\pi\in\Pi} L(\pi,r) = \min_{r\in\mathcal{R}} \max_{\hat{\rho}\in\mathcal{C}_{\widehat{T}}} \underbrace{\alpha\widehat{H}(\pi) + Z_\beta\left( \mathbb{E}_{\hat{\rho}}\big[r(s,a)\big] - \mathbb{E}_{\tilde{\rho}^I}\big[r(s,a)\big] + \psi(r) \right)}_{\doteq \bar{L}(\hat{\rho},r)}. \qquad (13)$$

It is easy to see that $\mathcal{R}$ is compact and convex. Besides, from the proof of Ho & Ermon (2016, Proposition 3.1), $\mathcal{C}_{\widehat{T}}$ is also a compact and convex set. Accordingly, based on the concavity of $\bar{H}$ (Lemma 4.2), the minimax theorem holds (Du & Pardalos, 2013), and hence we have

$$
\begin{aligned}
\min_{r\in\mathcal{R}} \max_{\hat{\rho}\in\mathcal{C}_{\widehat{T}}} \bar{L}(\hat{\rho},r) &= \max_{\hat{\rho}\in\mathcal{C}_{\widehat{T}}} \min_{r\in\mathcal{R}} \bar{L}(\hat{\rho},r) \\
&= \max_{\hat{\rho}\in\mathcal{C}_{\widehat{T}}} \alpha\bar{H}(\hat{\rho}) + Z_\beta \left( \min_{r\in\mathcal{R}} \mathbb{E}_{\hat{\rho}}\big[r(s,a)\big] - \mathbb{E}_{\tilde{\rho}^I}\big[r(s,a)\big] + \psi(r) \right) \\
&= \max_{\hat{\rho}\in\mathcal{C}_{\widehat{T}}} \alpha\bar{H}(\hat{\rho}) + Z_\beta \psi^*\big(\tilde{\rho}^I - \tilde{\rho}\big) \qquad \text{(from the definition of convex conjugate)} \\
&= \max_{\hat{\rho}\in\mathcal{C}_{\widehat{T}}} \alpha\bar{H}(\hat{\rho}) + Z_\beta D_\psi\big(\tilde{\rho}, \tilde{\rho}^I\big). \qquad\qquad (14)
\end{aligned}
$$

Additionally, denote $r^*$ and $\hat{\rho}^*$ as

$$r^* \in \arg\min_{r\in\mathcal{R}} \max_{\hat{\rho}\in\mathcal{C}_{\widehat{T}}} \bar{L}(\hat{\rho},r), \quad \hat{\rho}^* \in \arg\max_{\hat{\rho}\in\mathcal{C}_{\widehat{T}}} \alpha\bar{H}(\hat{\rho}) + Z_\beta D_\psi\big(\tilde{\rho}, \tilde{\rho}^I\big). \qquad (15)$$

Due to Eq. (14), $(r^*, \hat{\rho}^*)$ is a saddle point of $\bar{L}$, and thus $\hat{\rho}^* \in \arg\max_{\hat{\rho}\in\mathcal{C}_{\widehat{T}}} \bar{L}(\hat{\rho}, r^*)$. By Lemma 4.1, it is easy to see that policy $\pi^*$ (that corresponds to $\hat{\rho}^*$) satisfies $\pi^* \in \arg\max_{\pi\in\Pi} L(\pi, r^*)$, thereby completing the proof.

### B.2  PROOF OF THEOREM 4.2

We present the following two lemmas before our main result.

**Lemma B.1.** *Denoting $p_1(x,y) = q_1(x)q_1(y|x)$ and $p_2(x,y) = q_2(x)q_2(y|x)$ as two joint distributions over finite spaces, we can bound the total variation distance (TVD) between $p_1$ and $p_2$ as*

$$D_{\text{TV}}(p_1, p_2) \leq E_{x\sim q_1(x)}\big[D_{\text{TV}}(q_1(\cdot|x), q_2(\cdot|x))\big] + D_{\text{TV}}(q_1, q_2). \qquad (16)$$

*Proof.* The proof is straight-forward:

$$
\begin{aligned}
D_{\text{TV}}(p_1, p_2) &= \frac{1}{2} \sum_{x,y} \big|p_1(x,y) - p_2(x,y)\big| \\
&= \frac{1}{2} \sum_{x,y} \big|q_1(x)q_1(y|x) - q_2(x)q_2(y|x)\big| \\
&= \frac{1}{2} \sum_{x,y} \big|q_1(x)q_1(y|x) - q_1(x)q_2(y|x) + q_1(x)q_2(y|x) - q_2(x)q_2(y|x)\big| \\
&\leq \frac{1}{2} \sum_{x,y} q_1(x)\big|q_1(y|x) - q_2(y|x)\big| + \frac{1}{2} \sum_{x,y} q_2(y|x)\big|q_1(x) - q_2(x)\big|
\end{aligned}
$$

$$= \sum_x q_1(x) \cdot \frac{1}{2} \sum_y |q_1(y|x) - q_2(y|x)| + \frac{1}{2} \sum_x |q_1(x) - q_2(x)| \sum_y q_2(y|x)$$

$$= E_{x \sim q_1(x)} \left[ D_{\text{TV}}(q_1(y|x), q_2(y|x)) \right] + D_{\text{TV}}(q_1, q_2), \tag{17}$$

where the last equality is obtained due to $\sum_y q_2(y|x) = 1$. $\qquad \square$

**Lemma B.2.** *Suppose that we have two Markov chain transition distributions $T_1(s'|s)$ and $T_2(s'|s)$, and the initial state distributions are the same. Then, for each $h \in [1, 2, \dots)$, the TVD of state marginals in time step $h$ is bounded as*

$$D_{\text{TV}}(p_1^h, p_2^h) \le \sum_{h'=0}^{h-1} E_{s \sim p_2^{h'}} \left[ D_{\text{TV}} \left( T_1(\cdot|s), T_2(\cdot|s) \right) \right], \tag{18}$$

*where $p_i^h(s) \doteq \Pr(s_h = s \mid T_i, \mu)$ for $i = 1, 2$.*

*Proof.* First, we have

$$\left| p_1^h(s) - p_2^h(s) \right|$$

$$= \left| \sum_{s'} T_1(s|s') p_1^{h-1}(s') - \sum_{s'} T_2(s|s') p_2^{h-1}(s') \right|$$

$$\le \sum_{s'} \left| T_1(s|s') p_1^{h-1}(s') - T_2(s|s') p_2^{h-1}(s') \right|$$

$$= \sum_{s'} \left| T_1(s|s') p_1^{h-1}(s') - T_1(s|s') p_2^{h-1}(s') + T_1(s|s') p_2^{h-1}(s') - T_2(s|s') p_2^{h-1}(s') \right|$$

$$\le \sum_{s'} \left( T_1(s|s') \left| p_1^{h-1}(s') - p_2^{h-1}(s') \right| + p_2^{h-1}(s') \left| T_1(s|s') - T_2(s|s') \right| \right)$$

$$= \sum_{s'} T_1(s|s') \left| p_1^{h-1}(s') - p_2^{h-1}(s') \right| + E_{s' \sim p_2^{h-1}} \left[ \left| T_1(s|s') - T_2(s|s') \right| \right]. \tag{19}$$

Thus, we can write

$$D_{\text{TV}}(p_1^h, p_2^h)$$

$$= \frac{1}{2} \sum_s \left| p_1^h(s) - p_2^h(s) \right|$$

$$\le \frac{1}{2} \sum_s E_{s' \sim p_2^{h-1}} \left[ \left| T_1(s|s') - T_2(s|s') \right| \right] + \frac{1}{2} \sum_s \sum_{s'} T_1(s|s') \left| p_1^{h-1}(s') - p_2^{h-1}(s') \right|$$

$$\text{(using Eq. (19))}$$

$$= \frac{1}{2} \sum_s E_{s' \sim p_2^{h-1}} \left[ \left| T_1(s|s') - T_2(s|s') \right| \right] + \frac{1}{2} \sum_{s'} \left| p_1^{h-1}(s') - p_2^{h-1}(s') \right| \sum_s T_1(s|s')$$

$$= E_{s' \sim p_2^{h-1}} \left[ \frac{1}{2} \sum_s \left| T_1(s|s') - T_2(s|s') \right| \right] + \frac{1}{2} \sum_{s'} \left| p_1^{h-1}(s') - p_2^{h-1}(s') \right|$$

$$\text{(using } \sum_s T_1(s|s') = 1\text{)}$$

$$= E_{s' \sim p_2^{h-1}} \left[ D_{\text{TV}} \left( T_1(\cdot|s'), T_2(\cdot|s') \right) \right] + D_{\text{TV}}(p_1^{h-1}, p_2^{h-1}) \tag{20}$$

$$\le \sum_{h'=0}^{h-1} E_{s \sim p_2^{h'}} \left[ D_{\text{TV}} \left( T_1(\cdot|s), T_2(\cdot|s) \right) \right] + D_{\text{TV}}(p_1^0, p_2^0) \qquad \text{(iteratively using Eq. (20))}$$

$$= \sum_{h'=0}^{h-1} E_{s \sim p_2^{h'}} \left[ D_{\text{TV}} \left( T_1(\cdot|s), T_2(\cdot|s) \right) \right], \qquad \text{(due to same initial state distributions)}$$

which completes the proof. $\qquad \square$

Observe that Lemma B.1 bounds the TVD of a joint distribution by the TVDs of its corresponding conditional and marginal distributions, and that Lemma B.2 bounds the difference of two MDPs' state visitations in each time step by the cumulative dynamics differences. Next, we provide the following lemma that bounds the difference between the expert's and learned policy's occupancy measures from above.

**Lemma B.3.** *For each $\hat{\rho} \in \mathcal{C}_{\widehat{T}}$, denote $\hat{\pi}$ as its corresponding stationary policy, i.e., $\hat{\pi} \doteq \hat{\rho}(s,a)/\sum_{a'} \hat{\rho}(s,a')$, and $\rho^{\hat{\pi}}$ denote the occupancy measure of $\hat{\pi}$ under true transition dynamics $T$. Then, the following holds:*

$$D_{\mathrm{TV}}(\rho^{\hat{\pi}}, \rho^E) \leq \frac{\gamma}{1-\gamma} \mathbb{E}_{s,a \sim \hat{\rho}} \Big[ D_{\mathrm{TV}}\big(T(\cdot|s,a), \widehat{T}(\cdot|s,a)\big) \Big] + D_{\mathrm{TV}}(\hat{\rho}, \tilde{\rho}^E) + D_{\mathrm{TV}}(\tilde{\rho}^E, \rho^E), \quad (21)$$

*where $\rho^E$ is the occupancy measure of expert policy $\pi^E$ under the true transition dynamics.*

*Proof.* For conciseness, let $\rho_1 \doteq \rho^{\hat{\pi}}$ and $\rho_2 \doteq \hat{\rho}$. Using the triangle inequality, it is easy to see that

$$D_{\mathrm{TV}}(\rho_1, \rho^E) \leq D_{\mathrm{TV}}(\rho_1, \rho_2) + D_{\mathrm{TV}}(\rho_2, \tilde{\rho}^E) + D_{\mathrm{TV}}(\tilde{\rho}^E, \rho^E), \quad (22)$$

where $\tilde{\rho}^E$ is the empirical occupancy measure of expert policy $\pi^E$. To bound $D_{\mathrm{TV}}(\rho_1, \rho_2)$, denoting $p_1^h(s,a) \doteq \mathrm{Pr}(s_h = s, a_h = a \mid T, \hat{\pi}, \mu)$ and $p_2^h(s,a) \doteq \mathrm{Pr}(s_h = s, a_h = a \mid \widehat{T}, \hat{\pi}, \mu)$ (the difference between them is marked in red), we can write

$$D_{\mathrm{TV}}(\rho_1, \rho_2) = \frac{1}{2} \sum_{s,a} \big| \rho_1(s,a) - \rho_2(s,a) \big|$$

$$= \frac{1}{2} \sum_{s,a} \left| (1-\gamma) \sum_{h=0}^{\infty} \gamma^h p_1^h(s,a) - (1-\gamma) \sum_{h=0}^{\infty} \gamma^h p_2^h(s,a) \right|$$

(using the definition of occupancy measure in Section 2)

$$= \frac{1-\gamma}{2} \sum_{s,a} \left| \sum_{h=0}^{\infty} \gamma^h \left( p_1^h(s,a) - p_2^h(s,a) \right) \right|$$

$$\leq \frac{1-\gamma}{2} \sum_{h=0}^{\infty} \sum_{s,a} \gamma^h \left| p_1^h(s,a) - p_2^h(s,a) \right|$$

$$= (1-\gamma) \sum_{h=0}^{\infty} \gamma^h \cdot \frac{1}{2} \sum_{s,a} \left| p_1^h(s,a) - p_2^h(s,a) \right|$$

$$= (1-\gamma) \sum_{h=0}^{\infty} \gamma^h \cdot D_{\mathrm{TV}} \left( p_1^h(s,a), p_2^h(s,a) \right)$$

$$\leq (1-\gamma) \sum_{h=0}^{\infty} \gamma^h D_{\mathrm{TV}} \left( p_1^h(s), p_2^h(s) \right), \quad (23)$$

where $p_1^h(s) \doteq \mathrm{Pr}(s_h = s \mid T, \hat{\pi}, \mu)$, $p_2^h(s) \doteq \mathrm{Pr}(s_h = s \mid \widehat{T}, \hat{\pi}, \mu)$, and the last inequation holds due to Lemma B.1 (note that $p_1^h(s,a) = p_1^h(s)\hat{\pi}(a|s)$ and $p_2^h(s,a) = p_2^h(s)\hat{\pi}(a|s)$).[4] Denote $T_1(s'|s) \doteq \sum_a T(s',a|s)$ and $T_2(s'|s) \doteq \sum_a \widehat{T}(s',a|s)$, where we slightly overload notations using $T(s',a|s) \doteq \hat{\pi}(a|s)T(s'|s,a)$ and $\widehat{T}(s',a|s) \doteq \hat{\pi}(a|s)\widehat{T}(s'|s,a)$. We obtain

$$D_{\mathrm{TV}} \left( T_1(\cdot|s), T_2(\cdot|s) \right) = \frac{1}{2} \sum_{s'} \left| T_1(s'|s) - T_2(s'|s) \right|$$

$$= \frac{1}{2} \sum_{s'} \left| \sum_a T(s',a|s) - \widehat{T}(s',a|s) \right|$$

---

[4]To avoid ambiguity, we use $D_{\mathrm{TV}}(p_1^h(s,a), p_2^h(s,a))$ and $D_{\mathrm{TV}}(p_1^h(s), p_2^h(s))$ to denote the TVDs between the corresponding state-action distributions and state distributions respectively.

$$\leq \frac{1}{2}\sum_{s',a}\left|T(s',a|s)-\widehat{T}(s',a|s)\right|$$

$$= D_{\text{TV}}\left(T(s',a|s),\widehat{T}(s',a|s)\right)$$

$$\leq E_{a\sim\hat{\pi}(a|s)}\left[D_{\text{TV}}\left(T(\cdot|s,a),\widehat{T}(\cdot|s,a)\right)\right].$$

(seeing $\hat{\pi}(a|s)$ as $q_1(x)$, $q_2(x)$, $T(s'|s,a)$ as $q_1(y|x)$, and $\widehat{T}(s'|s,a)$ as $q_2(y|x)$, and then using Lemma B.1)

Based on that, the following holds:

$$D_{\text{TV}}(\rho_1,\rho_2) \leq (1-\gamma)\sum_{h=0}^{\infty}\gamma^h D_{\text{TV}}\left(p_1^h(s),p_2^h(s)\right) \qquad\text{(from Eq. (23))}$$

$$\leq (1-\gamma)\sum_{h=1}^{\infty}\gamma^h \sum_{h'=0}^{h-1} E_{s\sim p_2^{h'}(s)}\left[D_{\text{TV}}\left(T_1(\cdot|s),T_2(\cdot|s)\right)\right]$$
$$\text{(using fact } D_{\text{TV}}(p_1^0(s),p_2^0(s)) = D_{\text{TV}}(\mu,\mu) = 0 \text{ and Lemma B.2)}$$

$$\leq (1-\gamma)\sum_{h=1}^{\infty}\gamma^h \sum_{h'=0}^{h-1} E_{s\sim p_2^{h'}(s)}\left[E_{a\sim\hat{\pi}(a|s)}\left[D_{\text{TV}}\left(T(\cdot|s,a),\widehat{T}(\cdot|s,a)\right)\right]\right]$$
$$\text{(using the above result)}$$

$$= (1-\gamma)\sum_{h=1}^{\infty}\gamma^h \sum_{h'=0}^{h-1} \mathbb{E}_{s,a\sim p_2^{h'}(s,a)}\left[D_{\text{TV}}\left(T(\cdot|s,a),\widehat{T}(\cdot|s,a)\right)\right]$$
$$\text{(noting that } p_2^h(s,a) = p_2^h(s)\hat{\pi}(a|s))$$

$$= (1-\gamma)\sum_{s,a} D_{\text{TV}}\left(T(\cdot|s,a),\widehat{T}(\cdot|s,a)\right)\sum_{h=1}^{\infty}\gamma^h \sum_{h'=0}^{h-1} p_2^{h'}(s,a)$$
$$\text{(expanding the expectation and rearranging terms)}$$

$$= \gamma(1-\gamma)\sum_{s,a} D_{\text{TV}}\left(T(\cdot|s,a),\widehat{T}(\cdot|s,a)\right)\underbrace{\sum_{h=1}^{\infty}\sum_{h'=0}^{h-1}\gamma^{h-1}p_2^{h'}(s,a)}_{\doteq A}$$

$$= \gamma(1-\gamma)\sum_{s,a} D_{\text{TV}}\left(T(\cdot|s,a),\widehat{T}(\cdot|s,a)\right)\underbrace{\sum_{h=1}^{\infty}\gamma^{h-1}\sum_{h'=0}^{\infty}\gamma^{h'}p_2^{h'}(s,a)}_{\doteq B}$$
$$\text{($B$ is derived by rearranging the terms in $A$)}$$

$$= \gamma(1-\gamma)\sum_{s,a} D_{\text{TV}}\left(T(\cdot|s,a),\widehat{T}(\cdot|s,a)\right)\sum_{h=1}^{\infty}\gamma^{h-1}\frac{\rho_2(s,a)}{1-\gamma}$$
$$\text{(noting that } \rho_2(s,a) = (1-\gamma)\sum_{h'=0}^{\infty}\gamma^{h'}p_2^{h'}(s,a))$$

$$= \sum_{s,a} D_{\text{TV}}\left(T(\cdot|s,a),\widehat{T}(\cdot|s,a)\right)\sum_{h=1}^{\infty}\gamma^h\rho_2(s,a)$$

$$= \sum_{s,a}\rho_2(s,a) D_{\text{TV}}\left(T(\cdot|s,a),\widehat{T}(\cdot|s,a)\right)\sum_{h=1}^{\infty}\gamma^h$$

$$= \frac{\gamma}{1-\gamma}\mathbb{E}_{s,a\sim\rho_2}\left[D_{\text{TV}}\left(T(\cdot|s,a),\widehat{T}(\cdot|s,a)\right)\right]. \qquad (24)$$

Substituting Eq. (24) in Eq. (22) gives the desired result. $\qquad\square$

Denoting $\rho^{\pi}$ as the occupancy measure of $\pi$ under underlying dynamics model $T$, We can write

$$J(\pi^E) - J(\rho^{\pi}) = \sum_{s,a}\rho^E(s,a)R(s,a) - \sum_{s,a}\rho^{\pi}(s,a)R(s,a) \qquad\text{(from the definition)}$$

$$
\begin{aligned}
&= \sum_{s,a} \left( \rho^E(s,a) - \rho^\pi(s,a) \right) R(s,a) \\
&\leq \sum_{s,a} \left| \rho^E(s,a) - \rho^\pi(s,a) \right| \qquad\qquad \text{(due to } |R(s,a)| \leq 1\text{)} \\
&= 2 D_{\text{TV}}(\rho^\pi, \rho^E).
\end{aligned}
\tag{25}
$$

Then, based on Lemma B.3, the desired result in Theorem 4.2 can be obtained by combining Eq. (25) with Eq. (21).

## B.3  PROOF OF THEOREM 4.3

Recall that $c(s,a) = C \cdot D_{\text{TV}}(T(\cdot|s,a), \widehat{T}(\cdot|s,a))$. We define

$$
\begin{aligned}
f(\rho) &\doteq \mathbb{E}_{s,a\sim\rho}[c(s,a)] + 2D_{\text{TV}}(\rho, \tilde\rho^E) \\
&= \sum_{s,a} c(s,a)\rho(s,a) + \left| \rho(s,a) - \tilde\rho^E(s,a) \right|.
\end{aligned}
\tag{26}
$$

Thanks to Lemma 4.1, minimizing the RHS of Eq. (7) is equivalent to the following problem:

$$
\min_{\rho \in \mathcal{P}(\mathcal{S}\times\mathcal{A})} f(\rho).
\tag{27}
$$

Let $\delta(s,a) \doteq \rho(s,a) - \tilde\rho^E(s,a)$. Then, Problem (27) can be transformed to the following one:

$$
\min_\delta \ \sum_{s,a} c(s,a)\delta(s,a) + \left| \delta(s,a) \right|
\tag{28}
$$

$$
\text{s.t.} \ \sum_{s,a} \delta(s,a) = 0
\tag{29}
$$

$$
\delta(s,a) \geq -\tilde\rho^E(s,a) \quad s \in \mathcal{S}, a \in \mathcal{A}.
\tag{30}
$$

For conciseness, we rewrite Problem (28)-(30) as the following form:

$$
\min_\delta \ g(\delta) \doteq \sum_{i=1}^n c_i\delta_i + |\delta_i|
\tag{31}
$$

$$
\text{s.t.} \ \sum_{i=1}^n \delta_i = 0
\tag{32}
$$

$$
\delta_i \geq -\tilde\rho_i^E \quad i \in [n]
\tag{33}
$$

where $i$ corresponds to a state-action pair, $n \doteq |\mathcal{S}| \cdot |\mathcal{A}|$, $[n] \doteq \{1, 2, \ldots, n\}$, and $\delta \doteq \{\delta_i : i \in [n]\}$.

Due to Eq. (32) and Eq. (33), $[n]$ can be divided into two disjoint sets, $\mathcal{N}_1(\delta) \doteq \{i \in [n] : \delta_i > 0\}$ and $\mathcal{N}_2(\delta) \doteq \{i \in [n] : \delta_i \leq 0\}$ ($\mathcal{N}_1(\delta) = \emptyset$ iff all $\delta_i = 0$). Thus, we can write

$$
g(\delta) = \sum_{i\in\mathcal{N}_1(\delta)} (c_i + 1)\delta_i + \sum_{j\in\mathcal{N}_2(\delta)} (c_j - 1)\delta_j.
\tag{34}
$$

For any $\delta$ meeting Constraints (32) and (33), we denote $\delta'$ (which should be $\delta'_{\mathcal{N}_1}$ if written in full) satisfying $\delta'_j = -\mathbf{1}[c_j - c_1^{\min} > 2] \cdot \tilde\rho_j^E$ for all $j \in \mathcal{N}_2(\delta)$, $\delta'_i = 0$ for all $i \in \mathcal{N}_1(\delta)\backslash\mathcal{N}_{\min}$, and $\delta'_i = \sum_{j\in\mathcal{N}_2(\delta)} \mathbf{1}[c_j - c_1^{\min} > 2] \cdot \tilde\rho_j^E / |\mathcal{N}_{\min}(\delta)|$ for all $i \in \mathcal{N}_{\min}(\delta)$, where $\mathcal{N}_{\min}(\delta) \doteq \{i \in \mathcal{N}_1(\delta) : i \in \arg\min_{i'\in\mathcal{N}_1(\delta)} c_{i'}\}$ and $c_1^{\min} \doteq \min_{i\in\mathcal{N}_1(\delta)} c_i$. Then, we have

$$
\begin{aligned}
g(\delta') &= \sum_{i\in\mathcal{N}_1(\delta)} (c_i + 1)\delta'_i + \sum_{j\in\mathcal{N}_2(\delta)} (c_j - 1)\delta'_j \\
&= \sum_{i\in\mathcal{N}_{\min}(\delta)} (c_i + 1)\delta'_i + \sum_{i'\in\mathcal{N}_1(\delta)\backslash\mathcal{N}_{\min}(\delta)} (c_{i'} + 1)\delta'_{i'} + \sum_{j\in\mathcal{N}_2(\delta)} (c_j - 1)\delta'_j \\
&= \sum_{i\in\mathcal{N}_{\min}(\delta)} (c_i + 1)\delta'_i + \sum_{i'\in\mathcal{N}_1(\delta)\backslash\mathcal{N}_{\min}(\delta)} (c_{i'} + 1)\delta'_{i'} - \sum_{j\in\mathcal{N}_2(\delta)} \mathbf{1}[c_j - c_1^{\min} > 2] \cdot (c_j - 1)\tilde\rho_j^E
\end{aligned}
$$

$$= (c_1^{\min} + 1) \sum_{j \in \mathcal{N}_2(\delta)} \mathbf{1}[c_j - c_1^{\min} > 2] \cdot \tilde{\rho}_j^E - \sum_{j \in \mathcal{N}_2(\delta)} \mathbf{1}[c_j - c_1^{\min} > 2] \cdot (c_j - 1) \cdot \tilde{\rho}_j^E$$

(due to $\delta_{i'}' = 0$)

$$= \sum_{j \in \mathcal{N}_2(\delta)} \mathbf{1}[c_j - c_1^{\min} > 2] \cdot (c_1^{\min} - c_j + 2) \cdot \tilde{\rho}_j^E. \tag{35}$$

Regarding $g(\delta)$, the following holds:

$$
\begin{aligned}
g(\delta) =& \sum_{i \in \mathcal{N}_1(\delta)} (c_i + 1)\delta_i + \sum_{j \in \mathcal{N}_2(\delta)} (c_j - 1)\delta_j \\
\geq& \sum_{i \in \mathcal{N}_1(\delta)} (c_1^{\min} + 1)\delta_i + \sum_{j \in \mathcal{N}_2(\delta)} \mathbf{1}[c_j - c_1^{\min} > 2] \cdot (c_j - 1)\delta_j \\
&+ \sum_{j \in \mathcal{N}_2(\delta)} \mathbf{1}[c_j - c_1^{\min} \leq 2] \cdot (c_j - 1)\delta_j \\
=& -\sum_{j \in \mathcal{N}_2(\delta)} \mathbf{1}[c_j - c_1^{\min} > 2] \cdot (c_1^{\min} + 1)\delta_j + \sum_{j \in \mathcal{N}_2(\delta)} \mathbf{1}[c_j - c_1^{\min} > 2] \cdot (c_j - 1)\delta_j \\
&+ (c_1^{\min} + 1) \left( \sum_{i \in \mathcal{N}_1(\delta)} \delta_i + \sum_{j \in \mathcal{N}_2(\delta)} \mathbf{1}[c_j - c_1^{\min} > 2] \cdot \delta_j \right) \\
&+ \sum_{j \in \mathcal{N}_2(\delta)} \mathbf{1}[c_j - c_1^{\min} \leq 2] \cdot (c_j - 1)\delta_j \\
& \qquad \text{(adding and subtracting } -\sum_{j \in \mathcal{N}_2(\delta)} \mathbf{1}[c_j - c_1^{\min} > 2] \cdot (c_1^{\min} + 1)\delta_j) \\
=& \sum_{j \in \mathcal{N}_2(\delta)} \mathbf{1}[c_j - c_1^{\min} > 2] \cdot (c_j - c_1^{\min} - 2)\delta_j \\
&+ (c_1^{\min} + 1) \left( \sum_{i \in \mathcal{N}_1(\delta)} \delta_i + \sum_{j \in \mathcal{N}_2(\delta)} \mathbf{1}[c_j - c_1^{\min} > 2] \cdot \delta_j \right) \\
&+ \sum_{j \in \mathcal{N}_2(\delta)} \mathbf{1}[c_j - c_1^{\min} \leq 2] \cdot (c_j - 1)\delta_j \\
\geq& \sum_{j \in \mathcal{N}_2(\delta)} \mathbf{1}[c_j - c_1^{\min} > 2] \cdot (c_1^{\min} - c_j + 2)\tilde{\rho}_j^E \\
&+ (c_1^{\min} + 1) \left( \sum_{i \in \mathcal{N}_1(\delta)} \delta_i + \sum_{j \in \mathcal{N}_2(\delta)} \mathbf{1}[c_j - c_1^{\min} > 2] \cdot \delta_j \right) \\
&+ \sum_{j \in \mathcal{N}_2(\delta)} \mathbf{1}[c_j - c_1^{\min} \leq 2] \cdot (c^{\min} + 1)\delta_j \qquad\qquad \text{(noting that } \delta_j \leq 0) \\
=& \sum_{j \in \mathcal{N}_2(\delta)} \mathbf{1}[c_j - c_1^{\min} > 2] \cdot (c_1^{\min} - c_j + 2)\tilde{\rho}_j^E \\
&+ (c_1^{\min} + 1) \left( \sum_{i \in \mathcal{N}_1(\delta)} \delta_i + \sum_{j \in \mathcal{N}_2(\delta)} \left( \mathbf{1}[c_j - c_1^{\min} > 2] + \mathbf{1}[c_j - c_1^{\min} \leq 2] \right) \cdot \delta_j \right) \\
=& \sum_{j \in \mathcal{N}_2(\delta)} \mathbf{1}[c_j - c_1^{\min} > 2] \cdot (c_1^{\min} - c_j + 2)\tilde{\rho}_j^E + (c_1^{\min} + 1) \underbrace{\left( \sum_{i \in \mathcal{N}_1(\delta)} \delta_i + \sum_{j \in \mathcal{N}_2(\delta)} \delta_j \right)}_{=0}
\end{aligned}
$$

(due to Constraint (32))

$$= g(\delta'). $$

(due to Eq. (35))

Denoting $\mathcal{G} \doteq \{\delta \in \mathbb{R}^{|\mathcal{S}| \cdot |\mathcal{A}|} \text{ s.t. (32) and (33)}\}$, we have the following fact:

$$\delta'_{\mathcal{N}_1} = \arg \min_{\delta \in \mathcal{G}(\mathcal{N}_1)} g(\delta), \tag{36}$$

where $\mathcal{G}(\mathcal{N}_1) \doteq \{\delta \in \mathcal{G} : \mathcal{N}_1(\delta) = \mathcal{N}_1 \text{ and } \mathcal{N}_2(\delta) = [n] \backslash \mathcal{N}_1\}$. Due to Eq. (35), we have

$$\min_{\delta \in \mathcal{G}} g(\delta) = \min_{\mathcal{N}_1 \subset [n]} g(\delta'_{\mathcal{N}_1}) = \min_{\mathcal{N}_1 \subset [n]} \sum_{j \in [n] \backslash \mathcal{N}_1} \mathbf{1}[c_j - c_1^{\min} > 2] \cdot (c_1^{\min} - c_j + 2) \cdot \tilde{\rho}_j^E. \tag{37}$$

Let $c^{\min} \doteq \min_{i \in [n]} c_i$ and $\mathcal{N}_1^* \doteq \{i \in [n] : c_i = c^{\min}\}$. The following fact is true:

$$
\begin{aligned}
g(\delta'_{\mathcal{N}_1}) - g(\delta'_{\mathcal{N}_1^*}) &= \sum_{j \in [n] \backslash \mathcal{N}_1} \mathbf{1}[c_j - c_1^{\min} > 2] \cdot (c_1^{\min} - c_j + 2) \\
&\quad - \sum_{j' \in [n] \backslash \mathcal{N}_1^*} \mathbf{1}[c_{j'} - c^{\min} > 2] \cdot (c^{\min} - c_{j'} + 2) \\
&\geq \sum_{j \in [n] \backslash \mathcal{N}_1} \mathbf{1}[c_j - c_1^{\min} > 2] \cdot (c^{\min} - c_j + 2) \\
&\quad - \sum_{j' \in [n] \backslash \mathcal{N}_1^*} \mathbf{1}[c_{j'} - c^{\min} > 2] \cdot (c^{\min} - c_{j'} + 2) \qquad (\text{due to } c^{\min} \leq c_1^{\min}) \\
&\geq 0,
\end{aligned}
\tag{38}
$$

where the last inequality holds because $\{j \in [n]/\mathcal{N}_1 : c_j - c_1^{\min} > 2\}$ is a subset of $\{j' \in [n]/\mathcal{N}_1^* : c_{j'} - c^{\min} > 2\}$. Thus, $\delta^* \doteq \delta'_{\mathcal{N}_1^*} = \min_{\delta \in \mathcal{G}} g(\delta)$, and we can express $\delta^*$ as

$$\delta^*(s,a) = \begin{cases} \frac{\sum_{s',a'} \mathbf{1}[c(s',a') - c^{\min} > 2] \cdot \tilde{\rho}^E(s',a')}{|\mathcal{N}_{\min}|}, & \textit{if } c(s,a) \leq c^{\min} \\ -\tilde{\rho}^E(s,a), & \textit{if } c(s,a) > c^{\min} + 2 \\ 0, & \textit{otherwise} \end{cases} \tag{39}$$

where $\mathcal{N}_{\min} = \mathcal{N}_1^*$. Due to $\delta(s,a) + \tilde{\rho}^E(s,a) = \rho(s,a)$, we obtain the optimal solution of Problem (27) as follows:

$$\rho^*(s,a) = \begin{cases} \frac{\sum_{s',a'} \mathbf{1}[c(s',a') - c^{\min} > 2] \cdot \tilde{\rho}^E(s',a')}{|\mathcal{N}_{\min}|} + \tilde{\rho}^E(s,a), & \textit{if } c(s,a) \leq c^{\min} \\ 0, & \textit{if } c(s,a) > c^{\min} + 2 \\ \tilde{\rho}^E(s,a), & \textit{otherwise} \end{cases} \tag{40}$$

thereby completing the proof.

### B.4 PROOF OF COROLLARY 4.1

Because $c(s,a) > c^{\min}$ when $\tilde{\rho}^D(s,a) = 0$, if $\tilde{\rho}^D(s,a) = 0$, then $\rho^*(s,a) = 0$ holds. The desired result can be easily obtained by seeing $\beta^*(s,a)\tilde{\rho}^D(s,a)$ as $\delta^*(s,a)$ in the proof of Theorem 4.3.

### B.5 MINIMIZING A CHI-SQUARED DIVERGENCE

The $f$-divergence between two distributions $\rho_1$ and $\rho_2$ is defined as

$$D_f(\rho_1 \| \rho_2) = \mathbb{E}_{\rho_2} \left[ f \left( \frac{\rho_1}{\rho_2} \right) \right] = \sup_g \mathbb{E}_{X \sim \rho_1}[g(X)] - \mathbb{E}_{X \sim \rho_2} \left[ f^*(g(X)) \right] \tag{41}$$

where $f^*$ is the convex conjugate. The $\chi^2$-divergence is the $f$-divergence with $f(x) = (x-1)^2$ and $f^*(y) = \frac{y^2}{4} + y$, i.e.,

$$\chi^2(\rho_1, \rho_2) = \sup_g E_{X \sim \rho_1}[g(X)] - E_{X \sim \rho_2} \left[ \frac{g(X)^2}{4} + g(X) \right] \tag{42}$$

By interpreting $g = -r$ and $X = (s, a)$, the following holds:

$$\chi^2(\rho_1, \rho_2) = \sup_r E_{(s,a) \sim \rho_2} \left[ r(s, a) \right] - E_{(s,a) \sim \rho_1}[r(s, a)] - \frac{1}{4} E_{(s,a) \sim \rho_2} \left[ r(s, a)^2 \right] \qquad (43)$$

Thus, using a convex reward regularizer $\psi(r) = \frac{r^2}{4\delta}$ enables CLARE to minimize a $\chi^2$-divergence between the target policy and learned policy, i.e., $\max_{\hat{\rho} \in \mathcal{C}_{\hat{T}}} \alpha \bar{H}(\hat{\rho}) - Z_\beta \delta \cdot \chi^2(\hat{\rho}, \hat{\rho}^*)$.

