# OpenReview forum: "CLARE: Conservative Model-Based Reward Learning for Offline Inverse Reinforcement Learning"
_ICLR.cc/2023/Conference — ICLR 2023 poster_

### Official Review · Reviewer_5Qn6 · 2022-10-17

**Confidence:** 2
**Correctness:** 4
**Technical Novelty And Significance:** 3
**Empirical Novelty And Significance:** 3
**Recommendation:** 6

**Clarity, Quality, Novelty And Reproducibility:**

I think the paper is of good quality and clarity and should be easy to reproduce.
I am however not familiar with IRL literature so cannot comment on novelty, as well as the selection of baseline algorithms.

**Strength And Weaknesses:**

Strength:
The paper is well written and easy to follow. I in particular like the baseline MOMAX. This is the simplest way to use offline data in the IRL pipeline and comparing with this baseline makes the empirical study more convincing. The proposed algorithm outperforms baselines by a large margin.

Weakness:
1) It is not clear why the y-axis in Figure 2 means. I presume it is the performance of the learned policy, since the algorithm needs to alternatively learn \pi and r. But I do not understand why BC is included, since it is not a inverse RL algorithm at all.
2) Table 1 may benefit from having second order (e.g., variance) and averaged statistics (e.g., expectation) instead of just the highest score

**Summary Of The Paper:**

The paper proposes a model-based reward learning algorithm. The key idea is to smartly balance the use of the expert data and the low-quality data in learning the reward function. Theoretical analysis is provided to justify the weight for each low-quality data. Empirical study also verifies the efficacy of the proposed algorithms.

**Summary Of The Review:**

see above

---

> ### Author Response · Authors · 2022-11-13
> **Reply to Reviewer 5Qn6**
>
> Thank you for your detailed and constructive comments! Below are detailed responses to each comment:
>
> **Q1: It is not clear what the y-axis in Figure 2 means and why BC is included.**
>
> The reviewer's understanding is correct. The y-axis means the average score of the learned policy. While BC is not an inverse RL algorithm, its problem setting is the same as offline inverse RL, i.e., learning from offline data without ground-truth reinforcement signals. Thus, BC along with other offline imitation learning methods (e.g., ValueDICE (Kostrikov et al. 2019), EDM (Jarrett et al. 2020)) have been commonly used in evaluating offline inverse RL algorithms (Chang et al. 2021).
>
> **Q2: Table 1 may benefit from having second order (e.g., variance) and averaged statistics (e.g., expectation) instead of just the highest score.**
>
> Thank you for your helpful suggestion! We have added the second-order information for our method Sec. 6 for Table 1 (please see the red text). We'd like to point out that the results in Table 1 already contains the average scores of each algorithm.
>
> **Reference**
>
> (Kostrikov et al. 2019) [Imitation Learning via Off-Policy Distribution Matching](https://arxiv.org/pdf/1912.05032.pdf).
>
> (Jarrett et al. 2020) [Strictly batch imitation learning by energy-based distribution matching](https://proceedings.neurips.cc/paper/2020/file/524f141e189d2a00968c3d48cadd4159-Paper.pdf).
>
> (Chang et al. 2021) [Mitigating Covariate Shift in Imitation Learning via Offline Data With Partial Coverage](https://proceedings.neurips.cc/paper/2021/file/07d5938693cc3903b261e1a3844590ed-Paper.pdf).

---

### Official Review · Reviewer_CxsR · 2022-10-23

**Confidence:** 4
**Correctness:** 3
**Technical Novelty And Significance:** 4
**Empirical Novelty And Significance:** 4
**Recommendation:** 8

**Clarity, Quality, Novelty And Reproducibility:**

**Clarity and Quality**
The paper is written in good English and reads well. While the motivation of the paper is properly introduced and the meaning of the loss function of Eq (2) is explained, the connection between the two is not fully straightforward. Specifically, I understand the problem of reward extrapolation, but I cannot fully agree that this problem is addressed in Eq (2). In this sense, I have some questions:
1. Why does the visitation distribution $\tilde{\rho}^D$ consider the union of the datasets $\mathcal{D}_E$ and $\mathcal{D}_B$, when the dataset $\mathcal{D}_E$ is already considered in the distribution $\tilde{\rho}^E$?
2. From an intuitive perspective, the quality of the recovered reward function should depend on the behavioral policy $\pi^B$ used to collect dataset $\mathcal{D}_B$. Does this aspect emerge from the analysis?

**Related Works**
Since the paper considers the setting of the offline IRL, I suggest the authors to include in the "Related work" section the papers that address the "truly batch model-free IRL":

[1] Klein, Edouard, et al. "Inverse reinforcement learning through structured classification." Advances in neural information processing systems 25 (2012).

[2] Pirotta, Matteo, and Marcello Restelli. "Inverse reinforcement learning through policy gradient minimization." Thirtieth AAAI Conference on Artificial Intelligence. 2016.

[3] Ramponi, Giorgia, et al. "Truly batch model-free inverse reinforcement learning about multiple intentions." International Conference on Artificial Intelligence and Statistics. PMLR, 2020.

**Novelty**
Although the approach builds upon a well-known IRL algorithm (MaxEntIRL), the approach shows a notable novelty.

**Reproducibility**
The appendix provides sufficient details to reproduce the results.


**Details Of Ethics Concerns:**

None.

**Strength And Weaknesses:**

**Strengths**
- The paper addresses a very important issue of IRL
- The paper provides both empirical and theoretical contributions showing, in both cases, the benefits of the proposed approach

**Weaknesses**
- The rationale behind the loss function of Eq (2) could be explained better
- The "Related work" section should include additional papers (as detailed below)

**Summary Of The Paper:**

The paper addresses the problem of reward extrapolation in inverse reinforcement learning (IRL). The authors propose a modification of the MaxEntIRL to properly address the problem by leveraging an additional dataset collected by a behavioral policy and properly adjusting the loss function, introducing additional terms for assessing the reward function on the samples of this dataset. The loss function depends on a auxiliary function $\beta(s,a)$. A theoretical analysis is presented to show the guarantees of the approach as well as proposing a suitable value of $\beta(s,a)$. An experimental evaluation is presented showing the advantages of the proposed approach over several baselines on multiple Mujoco environments.

**Summary Of The Review:**

Overall, the paper provides a relevant contribution to the IRL literature. The reward extrapolation is a relevant problem in IRL and the approach proposed by the paper is novel and has shown to be effective from both the theoretical and empirical sides. I suggest the authors to expand the explanation of the loss function of Eq (2). My evaluation of the paper is positive.

---

> ### Author Response · Authors · 2022-11-13
> **Reply to Reviewer CxsR**
>
> Thank you for your valuable and detailed feedback! Below are detailed responses to each comment, and new comments on them are very welcome!
>
> ---
>
> ## Q1: The connection between the motivation and Eqn. (2)
>
> **Response:** Thank you for pointing this out. To explain why Eqn. (2) can tackle the reward extrapolation error, please recall that Eqn. (2) includes three main terms: *(i)* $Z_{\beta} \mathbb{E}\_{s,a\sim\hat{\rho}^{\pi}}[{r}(s,a)]$, *(ii)* $\mathbb{E}\_{s,a\sim\tilde{\rho}^E}[{r}(s,a)]$ , and *(iii)* $\mathbb{E}\_{s,a\sim{\tilde{\rho}^D}}[{{\beta}(s,a)}{r}(s,a)]$. By increasing the rewards in terms *(ii)* and *(iii)*, the learned reward function would in turn implicitly "penalize" the model-generated data that deviates from expert behaviors or has less data support, and thus performs conservatively on out-of-distribution state-action spaces.
>
> **Why does the visitation distribution $\tilde{\rho}^D$ consider the union of the datasets, $\mathcal{D}_E$ and $\mathcal{D}_B$, when the dataset $\mathcal{D}_E$ is already considered in the distribution $\tilde{\rho}^E$?**
>
> The underlying rationale is that CLARE is designed to encourage the policy to follow the expert behaviors that enjoys more data support. If removing $\mathcal{D}_E$ from the term *(ii)*, ClARE would neglect the uncertainty in the expert data, and the learned reward function may lead the policy to take risky actions. For example, if the expert has used each of $a_1$ and $a_2$ equal times at state $s$ but the model at $(s,a_1)$ is more accurate than that at $(s,a_2)$, it is better to assign higher reward for $(s,a_1)$ for safe offline policy search.
>
> ---
>
> ## Q2: The impact of the behavior dataset
>
> **Response:** Yes. From Theorem 4.2 and Corollary 4.1, the policy learned by CLARE satisfies ($\alpha=0$)
>
> $$J(\pi^E)-J(\pi^\mathrm{CLARE}) \le C\cdot \mathbb{E}_{s,a\sim\tilde{\rho}^E}\left[D_\mathrm{TV}\big(T(\cdot|s,a),\widehat{T}(\cdot|s,a)\big)\right]+2D_\mathrm{TV}(\tilde{\rho}^E,\rho^E)$$
>
> If the expert dataset enjoys a small sampling error ($D_\mathrm{TV}(\tilde{\rho}^E,\rho^E)$ is small), the coverage of the behavioral data on the expert dataset will dominate the performance.
>
> ---
>
> ## Q3: Related work
>
> **Response:** Thank you for pointing out all these related papers on inverse RL! We have added and discussed these papers in Sec. 6 of the revised draft (please see the red text).
>
> ---

---

> > ### Comment · Reviewer_CxsR · 2022-12-01
> > **Reply to authors' feedback**
> >
> > I thank the authors for the feedback and I apologize for my late reply. I think that my concerns have been appropriately addressed by the authors. Thus, I confirm my positive evaluation.

---

> > > ### Author Response · Authors · 2022-12-01
> > > **Many thanks for your further updates!**
> > >
> > > Thank you so much for further reviewing our response and confirming the evaluation!

---

### Official Review · Reviewer_yF3G · 2022-10-25

**Confidence:** 2
**Correctness:** 3
**Technical Novelty And Significance:** 3
**Empirical Novelty And Significance:** 3
**Recommendation:** 8

**Clarity, Quality, Novelty And Reproducibility:**

The paper is clearly written, high quality, and novel. I only have minor concerns about the reproducibility as it involves many moving parts, but as the authors have open-sourced their implementation, I think this should be helpful towards that end.

**Strength And Weaknesses:**

Strengths:
- Paper is well written - especially in tackling two related, but still distinct issues in offline IRL, it is easy to follow the contributions and their exposition is well done.
- Along the above point, the combination of different techniques to improve overall performance of the offline IRL algorithm is quite a reasonable one, i.e. combinging model-based conservative policy learning into one technique.
- Results are very strong and convincing on known benchmarks
- The pointwise weight parameters is a novel and interesting contribution.

Weaknesses:
- Why is the reward regularizer chosen as r^2. I could not find any further elaboration on this, and the rest of the section on regluarizers discusses the conventional policy regluarization as a KL divergence. Theorem 4.1 states the reasoning behind various reward regularizations, which I could just be overlooking as I am not familiar with the cited works, however, it would be good to hear more about the reasons r^2 was chosen in terms of why it performs well with respect to the datasets used.
- As far as I can tell, the generalization ability coming from the policy updates under the learned dynamics model is not a novel contribution. I think it would benefit this paper's visibility on the overall contribution if they reframed their approach as being only the conservative reward function learning which is compatible with an off-the-shelf offline RL method rather than a two step update algorithm.

**Summary Of The Paper:**

CLARE uses a learned dynamics model and conservative policy objective in order to tackle the problem of learning a reward function from a dataset of expert trajectories, i.e. "offline IRL". Their contribution tackles both the reward extrapolation problem in offline IRL as well as improves generalization of the learned policy. The first well-known issue is tackled using the conservative reward update and a notion of safety constrained policy update. Secondly, the generalization to unseen states is handled by learning a dynamics model. Additionally, the authors provide a theoretical framework for balancing the exploitation-exploration tradeoff with a derived upper bound on the return gap between the learned and expert policies.

**Summary Of The Review:**

Overall, this is a nice paper with empirical and theoretical contributions and I would lean towards acceptance.

---

> ### Author Response · Authors · 2022-11-13
> **Reply to Reviewer yF3F**
>
> Thank you for your appreciation of the contribution and novelty of this paper! Below are detailed responses to each comment:
>
> **Q1: Why is the reward regularizer chosen as $r^2$?**
>
> The reason is that using $\frac{1}{4\delta}r^2$ enables CLARE to minimize a scaled $\chi^2$-distance (a $f$-divergence with $f(x)=\delta(x-1)^2$) between the state-action distributions of the target policy and the learned policy, i.e., $\max_{\hat{\rho}\in\mathcal{C}_{\widehat{T}}}\alpha\bar{{H}}(\hat{\rho}) - Z_\beta \delta\cdot\chi^2(\hat{\rho},\hat{\rho}^*)$. For clarification, we have added a proof in Appendix B.5.
>
> **Q2: It would benefit this paper's visibility on the overall contribution if they reframed their approach as being only the conservative reward function learning.**
>
> Thank you for your insightful suggestion! We have carefully revised the manuscript to highlight the conservative reward learning. More Specifically,  in the Introduction we have updated the sentences *"CLARE addresses the above-mentioned challenge..."* to *"CLARE addresses the above-mentioned challenge by appropriately integrating conservatism into the learned reward to alleviate the possible misguidance in out-of-distribution states, and improves the reward generalization ability by utilizing a learned dynamics model."*

---

> > ### Comment · Reviewer_yF3G · 2022-12-01
> > **Thank you for the response**
> >
> > Thank you to the authors for the detailed response and overall discussion to the reviewers. After reading, I am happy to maintain my score and believe this work would be of interest to the community.

---

> > > ### Author Response · Authors · 2022-12-01
> > > **Thank you**
> > >
> > > Thank you very much for further reviewing our response and the positive assessment of our paper!

---

### Official Review · Reviewer_Gh9M · 2022-10-30

**Confidence:** 3
**Correctness:** 2
**Technical Novelty And Significance:** 2
**Empirical Novelty And Significance:** 3
**Recommendation:** 6

**Clarity, Quality, Novelty And Reproducibility:**

Clarity is good but the novelty and original contributions are limited. Code is provided.

**Strength And Weaknesses:**

**Strengths:**

- The topic of IRL and issues related to covariate shift and generalization are important and of interest to the community.
- Empirical results show significant improvement over prior inverse offline RL methods.

**Weaknesses:**
- Novelty and contributions are limited. The benefit of combining expert and offline data via pessimism on rewards and model-learning is already introduced in prior work such as Chang et al. 2021. The multiplicative weights are also used by Yu et al. 2021. No new insights are revealed, e.g. the discussion is limited to the well-known covariate shift issue in offline learning, penalizing rewards of less visited state-actions pairs, etc.
- Theoretical results are very weak. An important issue is that while the paper focuses on issues such as covariate shift and generalization in high-dimensional and/or continuous environments, in several places state-action pair counts are used such as empirical data distributions based on counts (below Eq. (2)) and Assumption 4.1. These are not aligned with the high-dimensional problem considered where typically $|D(s,a)| = 0, 1$. Indeed, Assumption 4.1 can only hold in tabular using concentration inequalities. Furthermore, the statements are not very interpretable and do not give bounds on standard quantities such as suboptimality of learned policy with respect to target policy expressed in terms of sample size, data coverage, and hypothesis class cardinalities. Theory is also not very insightful e.g. Theorem 4.2 is very similar to Theorem 4.4 in Yu et al. 2020.
- I don't think it's meaningful to discuss exploration (e.g. in the abstract, Figure 1, Introduction, etc.) in the context of purely offline learning from datasets. Running iterations on a learned model is not exploration.
- Generally, the paper makes several strong claims that are not supported by either the analysis or experiments. For example, it is not correct to say that optimizing the upper bound leads to balancing exploration-exploitation. Or the findings do not show the algorithm "can effectively capture the expert intention" which is claimed in the paper. The claim in paragraph 3 of Section 3 is not well-supported "it is clear that utilizing a learned dynamics model is indeed beneficial due to its capability of providing broader generalization (compared to model-free counterparts)...". The question of generalization of model-based vs. model-free approaches is still debated and unclear.
- Empirical evaluations can be improved by providing ablation studies such as showing the impact of expert data sample size on performance, the impact of conservative reward weighting, and accessing the impact of offline data on performance. Finally, tests on environments with high-dimensional image inputs on Atari strengthen the work on the empirical aspects.




**References**

Chang, J., Uehara, M., Sreenivas, D., Kidambi, R., & Sun, W. (2021). Mitigating Covariate Shift in Imitation Learning via Offline Data With Partial Coverage. Advances in Neural Information Processing Systems, 34, 965-979.

Yu, T., Kumar, A., Rafailov, R., Rajeswaran, A., Levine, S., & Finn, C. (2021). Combo: Conservative offline model-based policy optimization. Advances in neural information processing systems, 34, 28954-28967.

Yu, T., Thomas, G., Yu, L., Ermon, S., Zou, J. Y., Levine, S., ... & Ma, T. (2020). Mopo: Model-based offline policy optimization. Advances in Neural Information Processing Systems, 33, 14129-14142.

**Summary Of The Paper:**

This paper studies the reward extrapolation challenge in offline inverse reinforcement learning. The setting considered is access to a dataset of expert demonstrations as well as an offline dataset, collected by a behavior policy that is not necessarily an expert policy. The authors propose a conservative IRL algorithm called CLARE that integrates conservatism into the learned reward function and exploits learned transitions. They provide some analysis of their approach and show improved empirical performance over prior methods on continuous control tasks.


**Summary Of The Review:**

Despite the improved empirical performance, I did not find algorithmic design and theoretical analysis to be significant. The paper discusses well-known insights and does not provide any new insights.

---

> ### Author Response · Authors · 2022-11-13
> **Reply to Reviewer Gh9M (3/3)**
>
> ## Q5: Empirical evaluations can be improved
>
> **Response:** Following the reviewer's suggestion, we have added ablation studies of the impacts of expert data sample sizes (Table 5 in Appendix A.3), reward weighting (Fig. 5 in Appendix A.3), accessing diverse data on performance (Table 4 in Appendix A.3) in the revision. Due to very time-consuming data collection needed for model structure design, it is infeasible to run the suggested experiment on Atari for this rebuttal, but we will continue to investigate.
>
> **References**
>
> (Ho et al. 2016) [Generative adversarial imitation learning](https://proceedings.neurips.cc/paper/2016/file/cc7e2b878868cbae992d1fb743995d8f-Paper.pdf).
>
> (Yu et al. 2020) [Mopo: Model-based offline policy optimization](https://proceedings.neurips.cc/paper/2020/file/a322852ce0df73e204b7e67cbbef0d0a-Paper.pdf).
>
> (Yu et al. 2021) [Combo: Conservative offline model-based policy optimization](https://proceedings.neurips.cc/paper/2021/file/f29a179746902e331572c483c45e5086-Paper.pdf).
>
> (Chang et al. 2021) [Mitigating Covariate Shift in Imitation Learning via Offline Data With Partial Coverage](https://proceedings.neurips.cc/paper/2021/file/07d5938693cc3903b261e1a3844590ed-Paper.pdf).
>
>  (Viano et al. 2021) [Robust inverse reinforcement learning under transition dynamics mismatch](https://proceedings.neurips.cc/paper/2021/file/d9e74f47610385b11e295eec4c58d473-Paper.pdf).
>
> (Garg et al. 2021) [IQ-Learn: Inverse soft-Q Learning for Imitation](https://proceedings.neurips.cc/paper/2021/file/210f760a89db30aa72ca258a3483cc7f-Paper.pdf)

---

> > ### Comment · Reviewer_Gh9M · 2022-12-08
> > **Thank you for your response**
> >
> > I thank the authors for their detailed response. The authors have addressed most of my concerns such as on comparison with the objective and performance of MILO, clarifying "exploration" in this setting, ablation studies, and improvements in the text. I still find the theoretical segment to be weak but I believe the main contribution of this work is the strong empirical results. Therefore I have increased my score to 6.

---

> > > ### Author Response · Authors · 2022-12-09
> > > **Thank you**
> > >
> > > Thank you so much for further reviewing our response and increasing the score!

---

> ### Author Response · Authors · 2022-11-13
> **Reply to Reviewer Gh9M (2/3)**
>
> ---
>
> ## Q2: Theoretical results
>
> **(1) The theoretical results are not aligned with the high-dimensional problem.**
>
> Analyzing high-dimensional and continuous spaces remains an open problem for offline RL/IRL. Along the same line as in many recent works (e.g., Ho et al. 2016, Viano et al. 2021, Garg et al. 2021), this work seeks to gain insights from the theoretical analysis of discrete spaces and then use it to guide the practical algorithm design for continuous environments. It is of great interest to investigate how to establish guarantees rigorously for continuous environments.
>
> In addition, we have sharpened our results, and it turns out that all main results do not need  Assumption 4.1, and this assumption is only used in the remark below Eqn. (7) to provide a quantified result for Eqn. (7) in terms of data sample size.  We have revised the paper accordingly.
>
> **(2) The statements are not very interpretable.**
>
> Per the reviewer's suggestion, we provide a more interpretable bounds in the following. From Theorem 4.2 and Corollary 4.1, it can be easily seen that for $\alpha=0$, the policy learned by CLARE satisfies
>
> $$J(\pi^E)-J(\pi^\mathrm{CLARE}) \le C\cdot \mathbb{E}_{s,a\sim\tilde{\rho}^E}\left[D\_\mathrm{TV}\big(T(\cdot|s,a),\widehat{T}(\cdot|s,a)\big)\right]+2D_\mathrm{TV}(\tilde{\rho}^E,\rho^E)$$
>
> By quantifying the TV distance via concentration assumptions, we can characterize the impact of sample sizes from this bound. Further, since the dynamics model is estimated on offline data, the bound also implies that with a good quality of the expert dataset ($D\_\mathrm{TV}(\tilde{\rho}^E,\rho^E)$ is small), the offline data coverage on empirical expert state-action traces would dominate the performance of CLARE.
>
> **(3) Theorem 4.2 is very similar to Theorem 4.4 in Yu et al. 2020.**
>
> Thank you for bringing this up, but we respectfully disagree with this comment. The two theorems provide distinct insights for offline forward RL and offline inverse RL, respectively. For comparison, we restate the result of Theorem 4.4 in Yu et al. 2020 as follows:
>
> $$\mbox{Theorem 4.4 (MOPO):}~J(\pi^E) - J(\pi^\mathrm{MOPO})\le 2\lambda\cdot \mathbb{E}\_{s,a\sim{\color{blue}\hat{\rho}^E}}\left[u(s,a)\right]$$
>
> $$\mbox{Theorem 4.2 (CLARE):}~J(\pi^E)-J(\pi) \le C\cdot \mathbb{E}\_{s,a\sim{\color{blue}\hat{\rho}^\pi}}\left[D_\mathrm{TV}\big(T(\cdot|s,a),\widehat{T}(\cdot|s,a)\big)\right]+ 2\left(D_\mathrm{TV}(\hat{\rho}^\pi,\tilde{\rho}^E) + D_\mathrm{TV}(\tilde{\rho}^E,\rho^E)\right)$$
>
> Theorem 4.4 in MOPO characterizes the return gap between the expert policy and ***the policy learned by MOPO*** in terms of the expected model error estimate over empirical expert data distribution.   In contrast, Theorem 4.2 in this paper characterizes the gap between the expert policy and ***a general policy*** in terms of *(i)* the expected model error over the policy's state-action distribution under learned dynamics, *(ii)* the difference of the state-action visitation between the policy and expert policy, and *(iii)* the sampling error of the expert dataset. Minimizing this bound can guide meaningful offline policy search in offline inverse RL.
>
> ---
>
> ## Q3: Exploration in this context
>
> **Response:** Sorry for the possible confusion. The "exploration" used in the context of this manuscript refers to searching for potential gains and enhancing the generalization ability of the algorithm, by escaping the offline data manifold via model rollout. For clarification, we have added quotation marks for both "exploitation" and "exploration" along with a footnote in the Introduction to highlight their specific meanings in the revision.
>
> ---
>
> ## Q4: Strong claims
>
> **(1) It is not correct to say that optimizing the upper bound leads to balancing exploration-exploitation.**
>
> The upper bound in Eq. (7) pinpoints to the "exploration-exploitation" tradeoff in this context (as illustrated in Sec. 4.2), and minimizing it needs to calibrate this balance appropriately.   We have revised the manuscript and changed these sentences to "appropriately balancing the tradeoff".
>
> **(2) The findings do not show the algorithm can "effectively capture the expert intention".**
>
> Since the policy search is guided by the learned reward function, the return gap between the expert and the learned policy indicates how far the learned reward is from the expert intention. The smaller the gap, the closer our method will capture the expert intention.
>
> **(3) The claim in paragraph 3 of Section 3 is not well-supported "it is clear that utilizing a learned dynamics model...".**
>
> Thank you for pointing this out. We have revised this sentence to: "Nevertheless, we believe that utilizing a learned dynamics model is beneficial because it is expected to provide broader generalization by learning on additional model-generated synthetic data."
>
> ---

---

> ### Author Response · Authors · 2022-11-13
> **Reply to Reviewer Gh9M (1/3)**
>
> ## Q1: Novelty and contributions
>
> **(1) The benefit of combining expert and offline data via pessimism on rewards and model-learning is already introduced in prior work such as Chang et al. 2021.**
>
> While both MILO (Chang et al. 2021) and our method seek to incorporate conservatism into the min-max inverse RL framework, there are some fundamental differences  in how to achieve that, which we elaborate in the following.
>
> $$\mbox{MILO:}~ \min_{r \in \mathcal{R}} \max_{\pi \in \Pi}\bigg(\mathbb{E}\_{(s, a) \sim {\color{blue}\hat{\rho}^\pi}}[\overbrace{r(s, a)+{\color{blue}b(s, a)}}^\text{error accumulation}]-\mathbb{E}\_{(s,a)\sim \tilde{\rho}^E}[r(s, a)]\bigg)\tag{1}$$
>
> $$\mbox{CLARE:}~\min_{{r}\in\mathcal{R}}\max_{\pi\in\Pi} \bigg(Z_{\beta} \mathbb{E}\_{(s,a)\sim\hat{\rho}^{\pi}}\big[{r}(s,a)\big] - \mathbb{E}\_{(s,a)\sim\color{blue}\tilde{\rho}^D}\big[{\color{blue}{\beta}(s,a)}{r}(s,a)\big]- \mathbb{E}\_{(s,a)\sim\tilde{\rho}^E}\big[{r}(s,a)\big]+Z_\beta\psi({r}) + \alpha\widehat{{H}}(\pi)\bigg)\tag{2}$$
>
> MILO uses a model inaccuracy estimate, $b(s,a)$, to ***explicitly penalize the reward on model-generated state-actions*** where the learned dynamics is untrustworthy, as in MOPO (Yu et al. 2020). However, both the reward and the estimate can be biased, which can induce large accumulated errors over time, and hence the performance of MILO strongly depends on the quality of $b(s,a)$. Unfortunately, it has been reported (please see Fig. 2 in Yu et al. 2021) that the existing model inaccuracy estimation techniques (e.g., uncertainty quantification) struggle to predict the true model inaccuracy when going further away from the data distribution. It would result in the unsatisfactory performance of MILO, where the model rollout easily generates largely out-of-distribution data.
>
> On the contrary, CLARE directly learns a conservative reward function by appropriately ***improving the reward weights on offline state-actions***, implicitly penalizing the out-of-distribution data with no need to integrate model inaccuracy measure for each model-generated data. Thus, CLARE can circumvent the error accumulation issue, achieving much better results. To corroborate this, we add additional comparison against MILO using 5,000 expert data and 200,000/500,000 medium data (similar to the setup in
> Chang et al. 2021), and the results are presented as follows.
>
> | Algorithm | Medium data | Ant | Halfcheetah | Hopper | Walker |
> | :---:  | :---:  | :---:  | :---: | :---: | :---: |
> | MILO | 200,000 | 2896 | 95 | 722 | 3060 |
> | CLARE | 200,000 | 3819 | 4929 | 2142 | 3625 |
> | MILO | 500,000 | 3075 | 238 | 1879 | 3327 |
> | CLARE | 500,000 | 3866 | 4978 | 2273 | 3632 |
>
> Clearly, CLARE  achieves higher scores over MILO. Due to the unreliable model inaccuracy estimate, MILO's performance degrades with a more limited offline data coverage. We run the experiment by tuning the open source code of MILO, but it does not work well in Halfcheetah in this setting.
>
> **(2) The multiplicative weights are also used by Yu et al. 2021.**
>
> The key step of COMBO (Yu et al. 2021) is
>
> $$\hat{Q}^{k+1} \leftarrow \arg \min\_Q {\color{blue}\beta}\left(\mathbb{E}\_{{s}, {a} \sim \rho({s}, {a})}[Q({s}, {a})]-\mathbb{E}\_{{s}, {a} \sim \mathcal{D}}[Q({s}, {a})]\right)+\frac{1}{2} \mathbb{E}\_{{s}, {a}, {s}^{\prime} \sim d_f}\left[\left(Q({s}, {a})-\widehat{\mathcal{B}}^\pi \hat{Q}^k({s}, {a})\right)^2\right]\tag{3}$$
>
> The multiplicative weight ($\beta$) used in COMBO is a uniform hyperparameter that balances the Bellman updates and conservative Q-function estimation. In contrast, the weights of CLARE ($\beta(s,a)$) are pointwise for each offline state-actions, and we derive their closed-form expressions to balance the two-tier tradeoff in offline model-based inverse RL. The two weights have distinct forms and are designed for different purposes, as can be seen clearly in (2) and (3).
>
> **(3) No new insights are revealed.**
>
> We respectfully disagree with the reviewer about the insights of this work. We summaize new insights as follows: 1) Our findings reveal that a subtle two-tier tradeoff between the "exploitation" (on both expert and diverse data) and the "exploration" (on learned dynamics model) needs to be carefully calibrated in offline model-based inverse RL; 2) we characterize this tradeoff by providing an upper bound on the return gap between the learned policy and the expert policy; 3) we introduce new pointwise weights for each offline state-action to strike this balance, and we derive its closed form to minimize the return gap. In particular, our method moves away the unreliable model inaccuracy estimate for model-generated data as in MILO, and thus ameliorates the error accumulation issue and enjoys better  efficiency.

---

> ### Author Response · Authors · 2022-11-15
> **Reply to Reviewer Gh9M (Summary of Changes)**
>
> Thanks for your detailed comments! Per your suggestions, we have made the following major revision: (1) added an explanation for the exploration in Sec. 1, (2) added ablation studies of reward weighting in Fig. 5 in Appendix A.3 and accessing diverse data in Table 4 in Appendix A.3, (3) added new experiments on the impact of expert data sample sizes in Table 5 in Appendix A.3, (4) sharpened the theoretical results in Corollary 4.1 in Sec. 4.2, (5) made various revisions throughout the paper based on all reviewers’ comments. These changes are highlighted with red-colored text. New comments on these changes are very welcome!

---

> ### Author Response · Authors · 2022-12-01
> **Followup Response to Reviewer Gh9M**
>
> We would like to thank you again for the time you dedicated to reviewing our paper! Since the end of the discussion is getting close and we have not heard back from you yet, please let us know if you have further questions on our response, and we will be more than happy to answer your questions.

---

### Decision · Program_Chairs · 2023-01-20

**Decision:**

Accept: poster

**Justification For Why Not Higher Score:**

The paper proposes a principled approach to an important problem in offline inverse reinforcement learning, taking one step beyond prior work. For a higher score, I would expect a more substantial contribution, such as a larger delta from previous work.

**Justification For Why Not Lower Score:**

A main concern was regarding the theoretical contribution. The reviewers and AC acknowledge that (inverse) RL theory that goes beyond the tabular / more limited setting is difficult and rare. In this context, having even limited theoretical insights is valuable. Rather than rejecting the paper, the authors are encouraged to carefully discuss limitations of their theoretical analysis.

**Metareview: Summary, Strengths And Weaknesses:**

The paper addresses the problem of offline inverse reinforcement learning, where learned reward estimates may be biased due to data mismatch. The paper proposes an approach based on conservatism and a careful balance between multi-level tradeoffs. The authors provide theoretical guarantees for the tabular setting and strong empirical results in the MuJoCo domain.

Reviewers highlighted the importance of the addressed problem, interesting approach and strong empirical results. At the same time, reviewers initially raised a number of concerns. Concerns ranged from needs for clarification to larger concerns about the novelty of the work and the significance of the theoretical results.

The authors responded to reviewer concerns during the rebuttal period. The responses have been well received, and clarification questions have been addressed. Remaining questions regarding novelty and the relative importance of the theoretical contribution were discussed among reviewers.

In the camera ready version of the paper, the authors are encouraged to emphasize the strong empirical results that their proposed method achieves. Regarding the theoretical results, the authors are encouraged to carefully discuss limitations, e.g., restriction to the tabular setting.

Overall, the paper addresses and important problem in an interesting and novel way, and achieves strong empirical results. On this basis, the paper is recommended for acceptance.

Noted typos:
- "continuous environments Jarrett et al. (2020)"
- "state-actions that characterized by larger"


**Note From Pc:**

if the above contains the word "oral" or "spotlight" please see: "oral" presentation means -> notable-top-5% and "spotlight" means -> notable-top-25%. As stated in our emails, we are disassociating presentation type from AC recommendations

**Summary Of Ac-Reviewer Meeting:**

The paper was discussed in an AC-reviewer meeting. At the start of the meeting each reviewer was asked in turn to summarize their present assessment of the paper given the reviewer response and other reviews. At this point, reviewer assessment of the paper already converged. The main positive argument was that this paper proposes a new approach to an important problem and achieves strong empirical results. A remaining concern was the significance of the theoretical analysis. At the end of the discussion the reviewers recommended acceptance with an emphasis on the empirial insights.